# Archaeogenetic analysis of Neolithic sheep from Anatolia suggests a complex demographic history since domestication

Erinç Yurtman[1,24], Onur Özer[1,2,24], Eren Yüncü[1,24], Nihan Dilşad Dağtaş[1], Dilek Koptekin [3], Yasin Gökhan Çakan [4], Mustafa Özkan[1], Ali Akbaba[5], Damla Kaptan[1], Gözde Atağ[1], Kıvılcım Başak Vural[1], Can Yümni Gündem[6], Louise Martin[7], Gülşah Merve Kılınç[1,8], Ayshin Ghalichi[1,9], Sinan Can Açan [1], Reyhan Yaka[1], Ekin Sağlıcan [1], Vendela Kempe Lagerholm[10], Maja Krzewińska[10], Torsten Günther[11], Pedro Morell Miranda[11], Evangelia Pişkin[12], Müge Şevketoğlu[13], C. Can Bilgin[1], Çiğdem Atakuman[12], Yılmaz Selim Erdal[14,15], Elif Sürer [16], N. Ezgi Altınışık[14,15], Johannes A. Lenstra [17], Sevgi Yorulmaz[1], Mohammad Foad Abazari [18], Javad Hoseinzadeh [19], Douglas Baird [20], Erhan Bıçakçı[4], Özlem Çevik[21], Fokke Gerritsen [22], Rana Özbal[23], Anders Götherström[10,25], Mehmet Somel [1,25✉], İnci Togan[1,25] & Füsun Özer [14,15,25✉]

Sheep were among the first domesticated animals, but their demographic history is little understood. Here we analyzed nuclear polymorphism and mitochondrial data (mtDNA) from ancient central and west Anatolian sheep dating from Epipaleolithic to late Neolithic, comparatively with modern-day breeds and central Asian Neolithic/Bronze Age sheep (OBI). Analyzing ancient nuclear data, we found that Anatolian Neolithic sheep (ANS) are genetically closest to present-day European breeds relative to Asian breeds, a conclusion supported by mtDNA haplogroup frequencies. In contrast, OBI showed higher genetic affinity to present-day Asian breeds. These results suggest that the east-west genetic structure observed in present-day breeds had already emerged by 6000 BCE, hinting at multiple sheep domestication episodes or early wild introgression in southwest Asia. Furthermore, we found that ANS are genetically distinct from all modern breeds. Our results suggest that European and Anatolian domestic sheep gene pools have been strongly remolded since the Neolithic.

[1] Department of Biological Sciences, Middle East Technical University, Ankara, Turkey. [2] Emmy Noether Group Evolutionary Immunogenomics, Max Planck Institute for Evolutionary Biology, Plön, Germany. [3] Department of Health Informatics, Middle East Technical University, Ankara, Turkey. [4] Department of Prehistory, Istanbul University, Laleli, Istanbul, Turkey. [5] Department of Anthropology, Ankara University, Ankara, Turkey. [6] Department of Archaeology, Batman University, Batman, Turkey. [7] Institute of Archaeology, University College London, London, UK. [8] Department of Bioinformatics, Graduate School of Health Sciences, Hacettepe University, Ankara, Turkey. [9] Department of Archaeogenetics, Max-Planck Institute for the Science of Human History, Jena, Germany. [10] Archaeological Research Laboratory, Department of Archaeology and Classical Studies, University of Stockholm, Stockholm, Sweden. [11] Department of Organismal Biology, Human Evolution Research Program, Uppsala University, Uppsala, Sweden. [12] Department of Settlement Archaeology, Middle East Technical University, Ankara, Turkey. [13] Centre for Archaeology, Cultural Heritage and Conservation, Cyprus International University, Nicosia, Cyprus. [14] Department of Anthropology, Hacettepe University, Ankara, Turkey. [15] Molecular Anthropology Group (Human_G), Hacettepe University, Ankara, Turkey. [16] Department of Modeling and Simulation, Graduate School of Informatics, Middle East Technical University, Ankara, Turkey. [17] Faculty of Veterinary Medicine, Utrecht University, Utrecht, Netherlands. [18] Research Center for Clinical Virology, Tehran University of Medical Sciences, Tehran, Iran. [19] Department of Archaeology, University of Kashan, Kashan, Iran. [20] Department of Archaeology, Classics, and Egyptology, University of Liverpool, Liverpool, UK. [21] Department of Archaeology, Trakya University, Edirne, Turkey. [22] Netherlands Institute in Turkey, Istanbul, Turkey. [23] Department of Archaeology and History of Art, Koç University, Istanbul, Turkey. [24]These authors contributed equally: Erinç Yurtman, Onur Özer, Eren Yüncü. [25]These authors jointly supervised this work: Anders Götherström, Mehmet Somel, İnci Togan, Füsun Özer. ✉email: msomel@metu.edu.tr; fusunozer@hacettepe.edu.tr

Sheep was one of the four main animal species managed and domesticated during the Neolithic transition in southwest Asia c.10000–8000 before common era (BCE)[1,2]. The wild ancestor is thought to have been the Asian mouflon (*Ovis orientalis*), which was, by the early Holocene, distributed from west Anatolia to east Zagros. Its close relatives Urial sheep (*Ovis urial*) and Argali sheep (*Ovis ammon*) are distributed to the east of this region (Supplementary Note 1).

Archaeological evidence indicates that sedentary human communities were practicing sheep management already by 9000–8000 BCE in an area ranging from central Turkey to northwest Iran[3–5]. This is evidenced, for instance, by signs of corralling in the central Anatolian site Aşıklı Höyük[6–8] and young male kill-off practices identified in southeast Anatolian Çayönü[9] and Nevali Çori[10,11]. After 7500 BCE, young male kill-off as well as domestication-related morphological changes, such as small size, became widespread across the Fertile Crescent (including central Anatolia), as in the 7th millennium central Anatolian site of Çatalhöyük[11]. Following 7000 BCE, along with other elements of Neolithic lifeways, humans spread domesticated sheep to neighboring regions, including Europe, north Africa, and central Asia[3–5].

Both zooarchaeological data[3] and genomic evidence[12,13] imply a complex demographic history of domestic sheep. One notable pattern involves the high levels of genetic heterogeneity in domestic sheep. This includes multiple distinct mitochondrial DNA haplogroups found in modern breeds[14], as well as high nuclear genetic diversity in sheep compared to that in some other domesticates, such as cattle or dog[12,15]. High diversity would be consistent with scenarios where domestication involved multiple centers and/or a large and heterogeneous wild population. A non-exclusive scenario would be major introgression from wild sheep into domestic flocks, a notion supported by zooarchaeological evidence[3].

Another notable observation is the genetic structure of domestic sheep. Present-day domestic sheep cluster in two main groups based on genome-wide polymorphism data: east (Asian and African, including east Mediterranean islands) and west (European)[12,15,16]. Similarly, Asian and European sheep tend to carry higher proportions of mitochondrial DNA haplogroups A and B, respectively[17–19] (Supplementary Note 1). Recent ancient DNA work suggests that this pattern may have been established in Asia already by the 7th millennium BCE[20], and by the 2nd millennium BCE[21,22] in Europe.

At the same time, genomic analyses suggest high degrees of allele sharing across domestic sheep breeds. This has been considered as evidence for the recent spread of sheep with desired traits across the globe, especially within the last five millennia, as part of the "Secondary Products Revolution"[13,23–25]. Although the first domesticated sheep were likely used for their meat[23] and possibly their milk[26], they started to be increasingly exploited for their wool in Bronze Age southwest Asia during the 4th and 3rd millennia BCE[23,27]. Intriguingly, a comparison of DNA retroelements across modern breeds implies an expansion of SW Asian lineages estimated to date back to the Bronze Age; according to this model, SW Asian sheep with desired traits, such as fine wool, were introduced into local breeds across the globe[28]. A recent ancient DNA study reports evidence consistent with novel breeds being introduced into Europe during the Bronze Age, coinciding with archaeological evidence for the introduction of wool to this continent[29]. In later periods, export and admixture of selected sheep breeds into local stocks continued, possibly across the globe[12]. Indeed, using haplotype sharing information, the most recent common ancestor of domestic sheep breeds has been inferred to date back only 800 generations ago (about 3200 years[30])—a result interpreted as evidence for a common origin of all domestic breeds[12].

We currently lack a solid demographic history model to explain these observations: high diversity, clear genetic structure, and high degrees of haplotype sharing. What is missing is genetic data on the initial steps of domestication and characterization of the early domesticated sheep gene pool. Here, we present the first attempt to bridge this gap by studying the ancient DNA of Neolithic period sheep remains from Anatolia, which partly includes the putative zone of sheep domestication[5]. Analyzing both nuclear polymorphism data and mitochondrial DNA (mtDNA) sequences in comparison with modern-day genomes as well as ancient sheep genomes from central Asia, we find support for the notions that the present-day domestic sheep population has complex origins, and that the western Eurasian sheep gene pool changed considerably since the Neolithic period.

## Results

We analyzed DNA from 180 archaeological sheep bone and tooth samples from late Pleistocene and early Holocene Anatolia, originating from six different sites from central and west Anatolia and spanning the Epipaleolithic/Pre-Pottery Neolithic ($n = 7$) and early to late Pottery Neolithic ($n = 173$) periods (Fig. 1 and Supplementary Data 1). We generated genome-wide ancient DNA data using shotgun sequencing and enrichment capture targeting single-nucleotide polymorphisms (SNP) from four samples, while mtDNA sequences were obtained and analyzed for 91 samples (Supplementary Data 1 and 2). We compared these data with published datasets[12,31] from present-day domestic sheep breeds and ancient Neolithic and Early Bronze Age (EBA) sheep (7012–4260 BCE) of Kyrgyzstan (Obishir V, abbreviated as "OBI")[20,32] (Fig. 1, Supplementary Fig. 1, and Supplementary Table 1).

**Ancient genome data collection, characterization, and authentication**. We prepared Illumina high-throughput sequencing libraries from 29 ancient sheep samples as well as negative controls from DNA extraction and library preparation steps (Supplementary Data 2). Quantifications of negative control libraries did not indicate any contamination. Four Anatolian Neolithic sheep (ANS) individuals' libraries containing >1% endogenous sheep DNA (median 2%) were chosen for further analyses. Three of these were from the central Anatolian site Tepecik-Çiftlik Höyük (TEP03, TEP62, TEP83) and one from the west Anatolian site Ulucak Höyük (ULU31). All four individuals were AMS C14 dated to the 7th–8th millennium BCE (Table 1).

To increase coverage per single-nucleotide polymorphism (SNP), we enriched the libraries of these four individuals using hybridization capture, targeting 20,000 SNPs (Supplementary Data 3) and sequenced deeper. The capture procedure increased the endogenous DNA proportion 1.5–4 times (Table 1 and Supplementary Data 2). All four libraries exhibited 20–42% postmortem damage (PMD) induced C to T transitions at 5' ends of reads, expected for authentic ancient molecules (Supplementary Fig. 2). After trimming 10 bp from either end of the reads to remove postmortem damage-induced transitions, we called SNPs from these four libraries using the Illumina OvineSNP50 Beadchip variant set[12,31], which included 40,136 SNPs mappable to the oviAri3 reference genome. This resulted in a dataset containing pseudohaploidised genotypes for c.6000–20000 autosomal SNPs per individual (Table 1). We further repeated the analyses using only transversion polymorphisms to avoid any residual postmortem damage effect (Supplementary Fig. 3; also see below). We also note that our dataset comprises SNPs mainly identified in modern breeds and is thus heavily influenced by SNP ascertainment bias, for which reason we chose to base our main

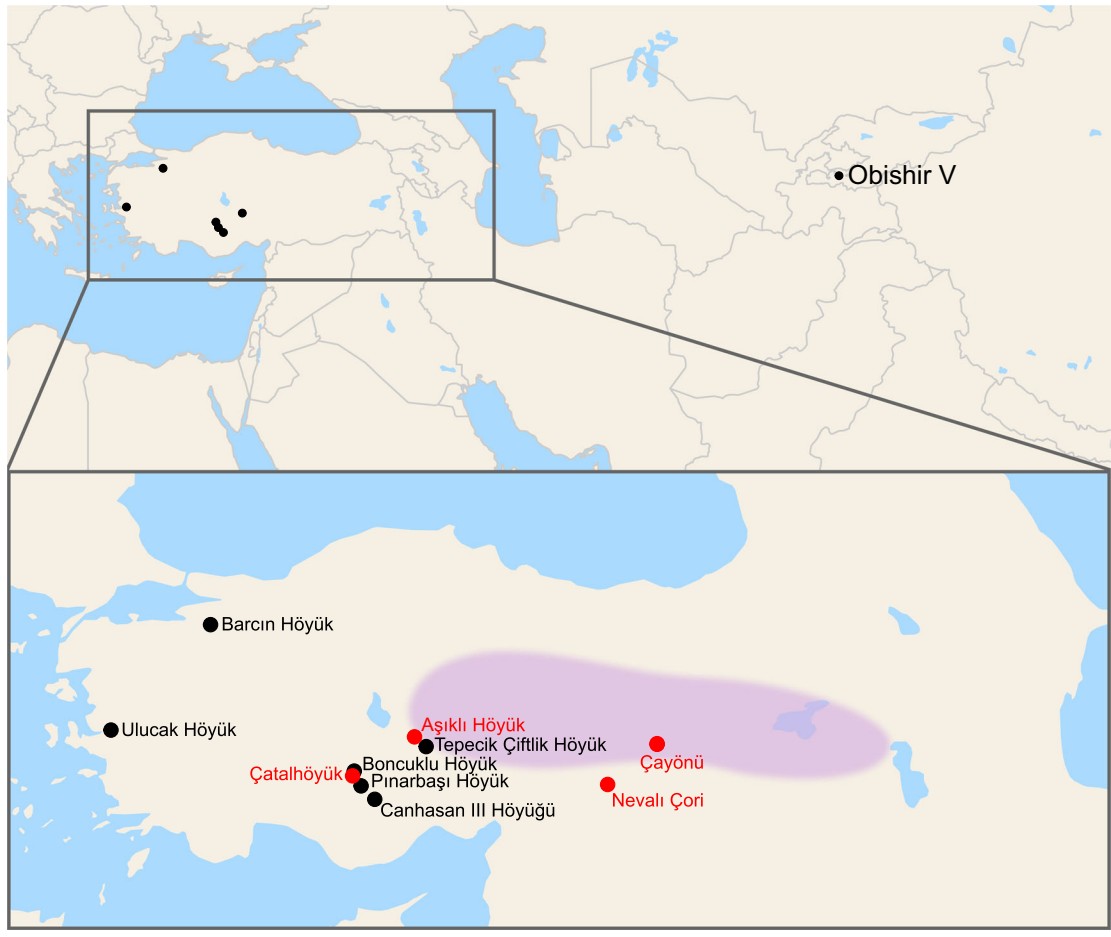

**Fig. 1 Map of archaeological sites.** Geographic origins of samples from six Anatolian archaeological sites studied in the present work (see Supplementary Note 2 for details) and one Kyrgyzstan archaeological site from Taylor et al.[20] are shown in black. Four additional Neolithic Anatolian sites relevant for sheep domestication discussed in the text are shown in red. The putative domestication center is depicted by the shaded area (adopted from Zeder[5]).

**Table 1 High-throughput sequencing summary statistics, AMS C14 ages, molecular sex estimates, and mtDNA haplogroups (HPGs) of four ancient sheep.**

| Sample ID | C14 ages cal. BCE | Genome coverage | Number of SNPs | Coverage per SNP | Molecular sex | mtDNA HPG |
|---|---|---|---|---|---|---|
| TEP62 | 7031–6687 | 0.276 | 20,281 | 3.89 | F | B |
| TEP03 | 7059–6756 | 0.103 | 15,794 | 2.32 | F | B |
| TEP83 | 6469–6361 | 0.023 | 6212 | 0.23 | M | A |
| ULU31 | 6227–6071 | 0.024 | 8533 | 0.44 | F | D |

"Genome coverage" values were calculated across the full genome using merged versions of the libraries shown in Supplementary Data 2. "Number of SNPs" indicate those that were covered by at least one read within a set of 40,136 SNPs, while "Coverage per SNP" indicates the mean number of reads across the same 40,136 SNPs after duplicate removal and quality filtering.

conclusions on *f*-statistics, which are known to be largely robust to ascertainment biases[33].

We noticed that the average fragment size for the three Tepecik-Çiftlik individuals was > 90 bp, uncommonly long for ancient DNA molecules (Supplementary Fig. 4). We used three approaches to investigate possible modern sheep DNA contamination in these libraries: (a) We selected DNA molecules bearing the postmortem damage signature (i.e., were most likely authentic) and repeated demographic analyses with only these plausibly authentic molecules. We found that using this restricted read set yields the same fundamental observations as using all reads (Supplementary Fig. 5). (b) We compared short and long molecules with respect to their postmortem damage signatures, which did not reveal any systematic difference (Supplementary Fig. 6). (c) We called genotypes using only

short and only long molecules (Supplementary Table 2). Using both datasets we then calculated outgroup $f_3$-statistics of the form $f_3(Argali; ANS_i, Modern)$, where $ANS_i$ represents one of the ANS individual's genotype based on either short or long reads, and *Modern* represents a modern sheep breed's genotype. We then calculated the Pearson correlation between outgroup $f_3$ values based on short vs. long reads, for each individual. The correlation coefficients were all positive (Spearman correlation coefficient $rho = [0.12–0.68]$) and nominally significant for TEP62 (Supplementary Fig. 7). None of the results indicated modern DNA contamination, leading us to conclude that the long DNA molecules of Tepecik–Çiftlik sheep most probably reflect unusual DNA preservation at this site, consistent with our earlier observations on Neolithic human material from Tepecik–Çiftlik[34].

We next sexed the four individuals. Given the lack of a sheep Y chromosome reference sequence, we compared autosomal versus X chromosomal coverages, which revealed three females and one male (Table 1 and Supplementary Fig. 8).

We also studied these four individuals' genotypes at 18 marker SNPs associated with putatively positively selected regions, identified by Kijas et al.[12] based on high genetic differentiation among modern-day breeds, including pigmentation, reproduction, and horn development. We found that among 16 loci where we could assign the ancestral state using *Ovis ammon* (Argali) and *Ovis vignei* (Urial) as outgroups, at six loci (38%), the derived allele was carried by at least one Anatolian Neolithic sheep individual, but never among all ANS (Supplementary Table 3). We caution, however, that the present-day linkage disequilibrium between marker alleles and the domestication phenotype-related causal alleles may not necessarily have existed 9000 years ago. Therefore, this result is not direct evidence that selection-related derived traits were already present in ANS.

**Anatolian Neolithic sheep show higher genomic affinity to modern European than non-European breeds.** To investigate sheep demographic history using our ancient genome dataset, we combined it with published genomic polymorphism data from ten present-day European [representing north, central, southwest (SW) and southeast (SE) Europe] and eight non-European breeds (from SW Asia, south (S) Asia, Africa, and east Mediterranean islands) generated on Illumina arrays[12,31] (Supplementary Fig. 1b and Supplementary Table 1). We also included data from three

Neolithic/EBA sheep partial genomes from Obishir V[20,32] (excluding the lowest coverage sample OB21-06/ENA Sample Accession No SAMEA802272), which included 636–1020 SNPs overlapping with the Illumina OvineSNP50 Beadchip 40k SNP set. In the analyses involving *f*-statistics, we used Argali sheep as the outgroup, after confirming that *D*-statistics of the form *D(Goat, Argali; Modern₁, Modern₂)* were all nonsignificant, which suggested that none of the modern breeds used here had received any visible Argali admixture (153 tests, multiple testing adjusted *P* > 0.05) (Supplementary Data 4 Sheet A).

We first summarized genome-wide variation through principal component analysis (PCA), calculating the principal components on the modern breed data and projecting the four ANS and three OBI genotypes onto the space described by the first two components (Fig. 2 and Supplementary Data 5; also see Supplementary Fig. 9 for the first and third components). As observed in earlier work, modern breeds of European descent and of non-European descent (Asian, African, and east Mediterranean islands) form two distinct clusters in the PCA. Within this worldwide constellation of modern sheep, ANS individuals attained a relatively central location, although conspicuously closer to the European cluster than to the non-European cluster. OBI sheep also inhabited a central location but unlike ANS, were closer to the non-European cluster. In parallel with the PCA results, ADMIXTURE[35] analysis of modern breeds and ANS indicated similarity between ancestry components in ANS and European breeds (OBI were not included in this analysis due to low SNP numbers). We observed that all four ANS individuals' ancestry component distributions showed a higher correlation with those of European breeds than non-European breeds (median Spearman rho ~0.95 vs. ~0.6,

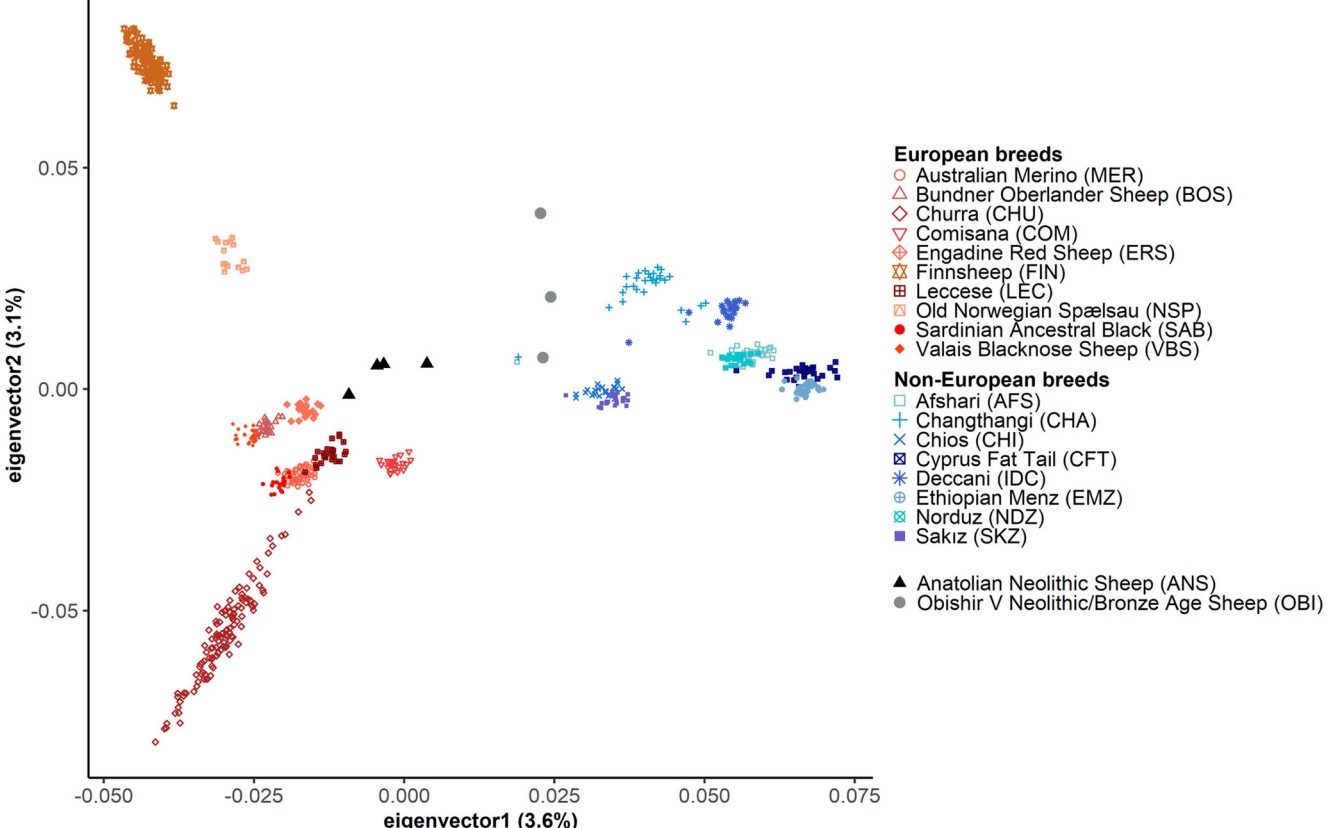

**Fig. 2 Genomic variation of modern breeds, Anatolian Neolithic (ANS) and Obishir V (OBI) sheep.** The graph represents the first two components of a PCA calculated using genotypes of 18 modern sheep breeds (*n* = 643). The four Anatolian Neolithic sheep individuals' genotypes (black triangles) and three Obishir V sheep individuals' genotypes (gray circles) were projected on these two components. Parentheses indicate the proportion of variance explained by the relevant component.

Mann–Whitney $U$ test $P < 0.02$) (Supplementary Figs. 10–12). Overall, PCA and ADMIXTURE-based analyses suggested that ANS show higher affinity to present-day European breeds.

To confirm these results within a formal statistical framework, we compared ANS genotype profiles with those of modern breeds with $D$-statistics[33]. We first calculated $D(Argali, ANS_1; ANS_2, Modern)$, where $ANS_1$ and $ANS_2$ denote two different ANS individuals and $Modern$ denotes any of the 18 present-day breeds. ANS were consistently genetically closer to each other than to present-day breeds (99% of 216 tests, multiple testing adjusted $P < 0.05$) (Supplementary Data 4 Sheet B). Meanwhile, tests of the form $D(Argali, Modern; ANS_1, ANS_2)$ showed that modern breeds did not show higher affinity to any ANS individual over any other ANS individual (108 tests, multiple testing adjusted $P > 0.05$) (Supplementary Data 4 Sheet C). Likewise, $D(Argali, ANS_1; ANS_2, ANS_3)$ showed no significant affinity between any pair of ANS relative to another individual (12 tests, multiple testing adjusted $P > 0.05$) (Supplementary Data 4 Sheet D). These nonsignificant results were not caused by lack of power, as >3000 SNPs were available in each comparison. We further used the outgroup $f_3$-statistics[33] to measure shared genetic drift between ANS and modern breeds. The distributions of $f_3(Argali; Modern, ANS)$ (Supplementary Fig. 13; Supplementary Data 6 Sheets A, B) were highly correlated between pairs of ANS individuals (Spearman correlation $rho = [0.50–0.87]$, $P < 0.05$; Supplementary Fig. 14), indicating that the affinities of ANS to modern breeds were highly alike, irrespective of ANS origin or historical age. These observations together suggest that ANS were genetically a relatively homogeneous population.

In agreement with the grouping between ANS and modern European breeds in the PCA, ANS genomes showed significantly higher $f_3$ values with those of European breeds (median = 0.184) than with non-European breeds (median = 0.176; Mann–Whitney $U$ test $P = 0.0005$) (Fig. 3a), indicating stronger affinity of ANS to present-day European breeds. We further examined this pattern by two approaches. Tests of the form $D (Argali, ANS; European, non-European)$ revealed that ANS had a higher affinity to European breeds, such that 50% of 80 tests were significant in this direction and no test was significant in the reverse direction (multiple testing adjusted $P < 0.05$) (Supplementary Data 4 Sheet E).

This was notable, given that the non-European breeds included east Mediterranean and SW Asian populations (CHI, CFT, NDZ, SKZ) that are geographically closest to the ANS individuals' provenance among all modern-day breeds analyzed.

To rule out the possibility that the observed patterns may be influenced by residual postmortem damage in ancient DNA that may not have been fully removed by read trimming, we repeated the aforementioned analyses using only transversion SNPs. Both the PCA (Supplementary Fig. 15a), $D$-statistics (Supplementary Data 7 Sheets A-F), and outgroup $f_3$-statistics (Supplementary Fig. 15b and Supplementary Data 8) calculated using this SNP subset consistently showed higher affinity of ANS to European than non-European modern-day breeds.

We next studied the OBI data using $f$-statistics. Given the low SNP numbers of these samples and their consistent locations observed in the PCA (also reported by Taylor et al.[20]), we pooled the three individuals' data. In outgroup $f_3$-statistics, OBI showed higher shared drift with non-European breeds (median = 0.245) than with European breeds (median = 0.234) although the difference was not significant (Mann–Whitney $U$ test $P = 0.102$) (Fig. 3b and Supplementary Data 6 Sheet C). We then compared $f_3(Argali; OBI, Modern)$ and $f_3(Argali; ANS, Modern)$ values, and found no correlation between them (Spearman correlation $rho = −0.16$, $P > 0.1$). Meanwhile, calculating $D(Argali, OBI; Modern_1, Modern_2)$, we observed that OBI have significantly higher affinity to the S Asian breeds in our dataset, Deccani and Changthangi (CHA and IDC), relative to European, African, and east Mediterranean breeds (12% of 153 comparisons, multiple testing adjusted $P < 0.05$) (Supplementary Data 4 Sheet F). These disparate genetic affinities of ANS and OBI suggest that genetic structure in the domestic sheep pool was already present in the early millennia after sheep domestication.

To further investigate the relationship of the ANS and OBI samples to present-day domestic sheep, we calculated $D(Argali, Modern; OBI, ANS)$. The two S Asian breeds (CHA and IDC) and one SW Asian (Norduz, NDZ) breed showed significant affinity to OBI over ANS in three out of 18 tests (multiple testing adjusted $P < 0.05$). Intriguingly, $D$-statistics involving the remaining 15 present-day breeds, including European breeds, were nonsignificant but all showed a trend of higher affinity to OBI over ANS

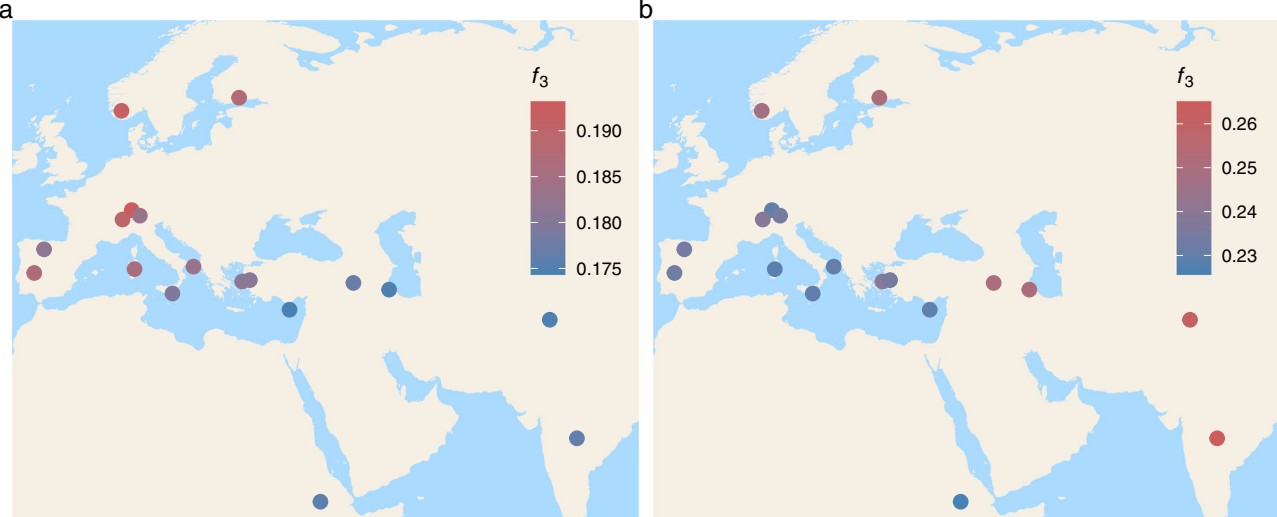

**Fig. 3 Genetic similarity between ANS, OBI, and present-day sheep populations.** Outgroup $f_3$-statistics were used to measure similarity between (**a**) ANS and present-day breeds, calculated as $f_3(Argali; ANS, Modern)$, using the joint allele frequencies of the four ANS individuals, (**b**) OBI and present-day breeds, calculated as $f_3(Argali; OBI, Modern)$, using the joint allele frequencies of the three OBI individuals. The dots indicate the locations of present-day breeds (Supplementary Table 1). Redder tones reflect higher $f_3$ values, which indicate higher shared drift (i.e., higher similarity) between ancient and modern-day sheep. See also Supplementary Fig. 13 for outgroup $f_3$-statistics calculated per ANS individual.

(Supplementary Data 4 Sheet G). This result would not be expected if ANS had been the direct ancestors of present-day European sheep.

We then calculated $D(Argali, Modern_1; Modern_2, ANS)$ and $D(Argali, Modern_1; Modern_2, OBI)$. In all comparisons performed as $D(Argali, Modern_1; Modern_2, ANS)$, modern breeds consistently grouped with other modern breeds over ANS (100% of 306 tests, multiple testing adjusted $P < 0.05$) (Supplementary Data 4 Sheet H). We observed a similar but weaker trend when we calculated $D(Argali, Modern_1; Modern_2, OBI)$, such that in 91% of comparisons modern breeds showed higher affinity to other modern breeds, at least as a trend (Supplementary Data 4 Sheet I). Finally, we tested whether ANS and OBI have higher affinity to each other or to modern breeds, calculating $D(Argali, OBI; ANS, Modern)$ and $D(Argali, ANS; OBI, Modern)$. Only c.1500 SNPs were available and none of the results were significant; still, we note that in all 18 tests ANS showed a trend of higher affinity to OBI than to modern breeds, but OBI showed higher affinity to modern breeds in 67% of the 18 tests (Supplementary Data 4 Sheets J, K). These results could have a number of explanations, including technical issues and complex demographic histories, which we discuss below.

**Mitochondrial DNA data supports the genetic affinity observed between ANS and modern European breeds**. The analyses above suggested that, within the present-day differentiation between European and non-European sheep, ANS were genetically closer to European breeds while OBI were closer to Asian breeds. Given the limited sample size for ANS genomes, we sought to replicate this pattern using mitochondrial DNA (mtDNA) data. mtDNA, although only informing about the maternal lineage, is more readily available across poorly preserved samples[36]. By collecting published mtDNA data, we confirmed that present-day breeds display strong geographic structure in their mtDNA haplogroup composition, with high frequencies of group A in central and east Asia, and high frequencies of group B in Europe and Anatolia, as reported in earlier work[17,18,21,37] (Supplementary Table 4). We note that neither haplogroup A nor B appears ancestral within sheep mitochondrial DNA diversity[14] (Supplementary Fig. 16 and Supplementary Data 9). All three OBI sheep carried haplogroup A[20]. We, therefore, hypothesized that this east-west differentiation in mtDNA haplogroup composition may have emerged already by the Neolithic period.

We amplified and Sanger sequenced a 144-bp fragment of the mtDNA control region, which contains five diagnostic marker SNPs for the five main haplogroups observed in present-day domestic sheep, i.e., haplogroups A-E[14], and is short enough to be analyzed in ancient samples[38,39] (Supplementary Table 5 and Supplementary Note 1). Using 311 modern mitogenomes, we estimated 95.5% accuracy of haplogroup assignment based on this fragment (Supplementary Fig. 16, Supplementary Data 9, 10, and Supplementary Note 1). Although misassignment can occur due to convergent transitions at diagnostic sites, a pattern we also identify among ancient sequences, we expect its overall occurrence to be low (Supplementary Note 1). We thus targeted the 144-bp fragment on DNA isolated from a total of 180 ancient sheep samples, each likely from distinct individuals based on their context (Supplementary Note 2). Negative controls included at each step of mtDNA analyses were free of contamination. We aimed to amplify at least two consistent sequences per sample to increase the likelihood of authenticity. Out of 91 sheep, 79 samples produced two or more sequences, whereas 12 samples yielded only a single sequence. The success rate of obtaining at least two sequences ranged between 22.5 and 60% across the six archaeological sites (Supplementary Table 6). The sequences thus obtained were analyzed to identify haplogroups and then compared across archaeological periods and regions (west

and central Anatolia) and with data from present-day Anatolian, European, and Asian sheep breeds (Fig. 4, Supplementary Tables 4, 7, and 8). We performed haplogroup comparisons at two levels: (i) using all ancient sequences, (ii) using a second higher confidence set from which we excluded single amplifications.

The estimated mtDNA haplogroup frequencies are presented in Fig. 4, which reveals a number of interesting patterns. First, we see the apparent dominance of haplogroup B in central and west Anatolia, from the Epipaleolithic to the late Neolithic (84%, 95% confidence interval: [76–92%]). This is in line with the hypothesis that the predominance of haplogroup B among present-day breeds from Europe (Supplementary Table 4), as well as west and central Turkey, may have been established by the Neolithic.

Second, we observe non-B haplogroups, including haplogroup A, in our ancient sample only post 7500 BCE, during the Ceramic period (Fig. 4). However, our sample size is too small to exclude the presence of non-B haplogroups in central Anatolia pre 7500 BCE. Whether there occurred an increase in haplogroup diversity in central and west Anatolia post-7,500 BCE yet remains unclear (Fig. 4 and Supplementary Note 1).

Finally, we observe clear changes in haplogroup composition between the Neolithic period and the present day. Most notably, haplogroup A, which occurs at a frequency of 5% in the pre-5500 BCE sample of 79 individuals (95% confidence interval: [1–12%]), reaches 26% in present-day central Anatolia, while remaining <5% in west Anatolia[38] (Supplementary Table 8). These analyses of maternal lineages suggest strong drift and/or admixture events in the post-Neolithic era sheep populations, at least in central Anatolia.

## Discussion

Our combined analyses of genome-wide polymorphism data and mitochondrial DNA from ancient central and west Anatolian sheep provide novel insights into sheep domestication and later dynamics. First, we find that Anatolian Neolithic sheep show significantly higher affinity to present-day European breeds than to Asian, African, and east Mediterranean breeds. This pattern is observed both with respect to genome-wide polymorphism, and also mitochondrial haplogroup composition, with mtDNA haplogroup B predominating in both ANS and present-day breeds from Europe. We further observe that Neolithic and Early Bronze Age sheep from southeastern Kyrgyzstan, compared to Anatolian Neolithic sheep, display a contrasting pattern of genetic affinity: ancient Kyrgyz sheep show greater affinity to present-day Asian sheep relative to European sheep and also carry the mtDNA haplogroup A.

These results suggest that the east-west genetic structure observed in modern breeds had already partly emerged by 7000–6000 BCE. We hypothesize that one section of the early domestic sheep gene pool, related to our ANS sample, was brought to Europe through the spread of Neolithic domesticates of the 7th and 6th millennia[40], whereas another section of the gene pool contributed to the OBI sample and modern-day Asian breeds.

One possible explanation for such an early differentiation pattern is the existence of multiple sheep domestication centers in the east and west of the Fertile Crescent, including central Anatolia. During the 9th to 8th millennia BCE, when evidence for sheep and goat management gradually accumulate in the zooarchaeological records[4,10,11], populations of local *Ovis orientalis* may have been domesticated by each region's Neolithic communities in parallel. The observed genetic structure in domestic sheep would then be reflecting the genetic structure of the local wild sheep involved. Indeed, zooarchaeological studies suggest that sheep and goat domestication likely involved multiple regions of the Fertile

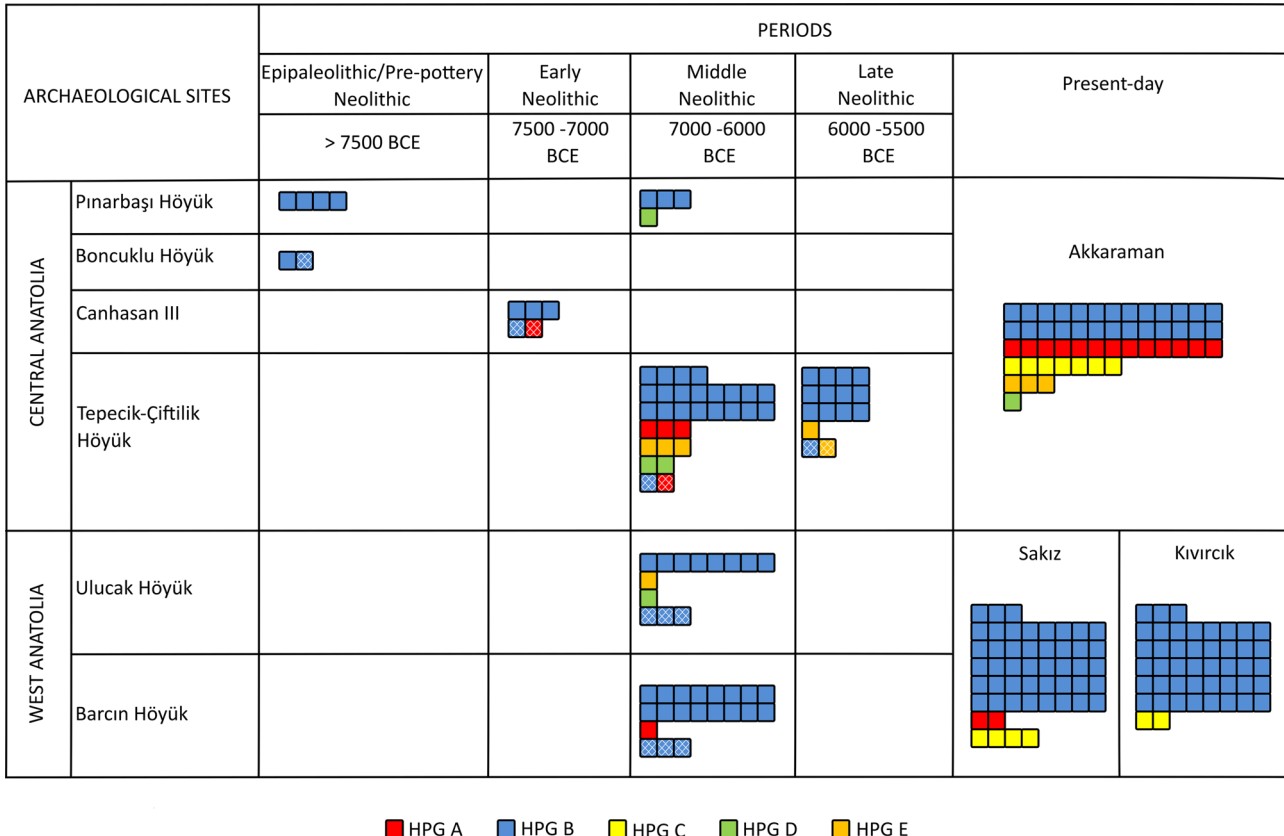

**Fig. 4 Change in mitochondrial lineages from Epipaleolithic to present-day in Anatolia.** Distribution of different haplogroups (HPG) (indicated by colors) of the studied sheep individuals. Color coding is as follows: HPG A (red), HPG B (blue), HPG C (yellow), HPG D (green), and HPG E (orange). The hatched pattern indicates individuals supported by a single sequence. Note that different periods and haplogroups are not homogeneously represented across different sites from a single region: e.g., >7500 BCE in central Anatolia is represented by Pınarbaşı and Boncuklu Höyük, while 7500–7000 BCE by Canhasan III. See Supplementary Data 1 for full details. For the samples with ages straddling two periods, we used their mean age to place them within an archaeological period. Haplogroup compositions of present-day breeds of Turkey were represented by Akkaraman (central Anatolia) and by Sakız and Kıvırcık (west Anatolia) (see Supplementary Table 4). Mitochondrial DNA haplotypes of ancient individuals are represented in Supplementary Fig. 17.

Crescent[4,41], a claim recently corroborated by ancient genomes in the case of goat domestication[42].

Alternative scenarios are also possible. For instance, the observed patterns could be explained by domestication of a single genetically diverse wild sheep population; following domestication this may have rapidly split into east and west populations accompanied by severe genetic bottlenecks, leading to the observed structure by the 7th millennium BCE. A single domestication event in the west (or east) Fertile Crescent, followed by rapid introgression by genetically differentiated wild sheep when early domestic populations spread to the east (or west), could also be plausible. Here it is interesting to note that in Anatolian sites dating to the early Chalcolithic (in Çatalhöyük West and in Erbaba in the Lakes district), more than one millennium after the initial decrease in sheep body size, sheep body sizes again rise to wild caprine levels, a pattern that has been interpreted as a sign of wild introgression[3]. Similar intogression patterns have been reported for pigs, cattle, and goats, supported by both zooarchaeological analysis[3] as well as ancient DNA[43,44]. Wild introgression into domestic sheep stocks could also have contributed to the observed differentiation patterns.

We note that determining the exact roles of these three non-exclusive scenarios, i.e. multiple domestication centers, rapid bottlenecks, and rapid introgression from the wild, will require further genetic sampling of ancient sheep, and in particular pre-domestic sheep populations in SW Asia.

A second, unexpected observation was that in D-statistics, present-day breeds were consistently closer to other present-day breeds than to ANS. This also included present-day European breeds and is inconsistent with a scenario where ANS would be the sole ancestral population of present-day European breeds.

One explanation for this could be technical biases differentiating our ANS data from other data. For instance, residual postmortem damage or the capture procedure could have introduced biases that artificially differentiate the ANS genotypes from those of modern-day breeds. However, when we test $D(Argali, Modern_1; Modern_2, ANS)$ using only transversion SNPs, we find the same result, i.e. modern-modern affinity to the exclusion of ANS (Supplementary Data 7 Sheet F). Further, we observe a similar trend in tests involving OBI, where data was produced in another laboratory using a protocol that removes postmortem-induced transitions, and without SNP capture. These observations lower the likelihood of a technical artifact causing the mentioned D-test results.

A biological explanation for these D-test results could be post-Neolithic gene flow events that influenced all sheep breeds, partly eclipsing the earlier genetic structure. For instance, a sheep lineage carrying a yet unsampled ancestry and bred for its fine wool may have spread during the Chalcolithic or Bronze Ages and dramatically influenced the global sheep gene pool, including European breeds. This scenario would be consistent with the aforementioned high degrees of haplotype sharing[12] or retrovirus

genotype sharing[28] observed among modern-day breeds, as well as recent ancient DNA work implicating post-Neolithic gene flow from eastern sources altering the west Eurasian sheep gene pools[29]. Hence, we hypothesize that central and west Anatolian Neolithic sheep were likely directly related to Europe's first domestic breeds, but the continent's domestic sheep gene pool was strongly remolded by subsequent gene flow events of Asian and/or African origin.

Finally, our results suggest that the east Mediterranean and Anatolian domestic sheep gene pools also underwent significant shifts since the Neolithic, and possibly of more dramatic magnitude than those in Europe. This can be deduced from the observation that Anatolian Neolithic sheep show higher genetic affinity to present-day European than to Anatolian breeds. This partly echoes what has been described for the human gene pool in Anatolia, such that Neolithic Anatolians show higher average similarity to present-day south Europeans than to present-day Anatolians[34,45].

## Methods

**Information about samples**. Ancient sheep bone and tooth samples ($n = 180$) were obtained from 6 archaeological sites in Anatolia: Pınarbaşı, Boncuklu Höyük, Tepecik-Çiftlik Höyük, and Canhasan III in central Anatolia; Ulucak Höyük and Barcın Höyük in west Anatolia (Fig. 1). Permits for archaeological samples processed in this study were obtained from relevant museums affiliated to the Republic of Turkey Ministry of Culture and Tourism. Brief information about the sites is provided in Supplementary Note 2. The whole-genome data published for two Neolithic OB20.06 (7012 +/− 45 BP) and OB20.01 (6989 +/−45 BP) and one Bronze Age sheep OB20.04 (4260 +/− 45 BP) from Obishir V, Kyrgyzstan[20,32] were also included the downstream analyses.

**AMS radiocarbon dating**. Five samples were AMS C14 dated at the TÜBİTAK-MAM (Gebze, Turkey) and one sample at Beta Analytic Inc. (London, UK) (Table 1). The IntCal13 calibration curve in OxCal v4.2 was used to calibrate radiocarbon ages estimated by TÜBİTAK-MAM and Beta Analytic Inc.[46,47]. The remaining samples were dated by the excavation directors based on their archaeological context.

**Procedures for contamination prevention and detection**. All laboratory analyses were carried out in a dedicated ancient DNA laboratory at the Middle East Technical University, Ankara, Turkey. Strict measures recommended for aDNA work to prevent contamination[48–50] were followed. All equipment and benchtops were frequently decontaminated with DNAaway (Molecular Bio-Products Inc.) or 2% bleach during the experiments. Negative controls were included at each step of the laboratory procedures. During DNA extraction two tubes of 80 mg hydroxyapatite were included as negative controls. The following controls were implemented to track contamination: one negative control was included at the beginning of all extraction sets to monitor modern contamination, and a second negative control was placed at the end of all sets to keep track of cross-contamination. In addition, the order of the samples was always kept the same throughout each extraction. For mtDNA amplification, PCR reactions were prepared in the aDNA laboratory and then immediately transferred to the post-PCR laboratory, which is located in a different building, to carry out amplification and agarose gel electrophoresis imaging. Two negative controls were included in each set of PCR amplification following the same rationale in the DNA extraction procedure. Negative controls were also included during the whole-genome library preparations. All of the negative controls were quantified with Agilent 2100 Bioanalyzer to assess contamination.

**Sample preparation and DNA extraction**. Prior to DNA extraction, the outer surfaces of bone samples were cleaned by abrasion with a cutting disk attached to a Dremel tool. A new cutting disk was used for each sample. DNA was extracted twice from each sample at different times following Dabney et al.[51]. A small piece of bone was cut out of cleaned bone and then ground to a fine powder with a mortar and pestle. In total, 120–200 mg of bone powder was weighed and placed in a 2-ml screw-top tube. Briefly, 1 ml extraction buffer (0.45 M EDTA and 0.25 mg/ml ProteinaseK) was added onto bone powder and incubated at 37 °C for 18 h in a rotator incubator After incubation, tubes were centrifuged and supernatants were transferred into tubes containing 13 ml binding buffer (M guanidine hydrochloride, 40% (vol/vol) isopropanol, 0.05% Tween-20, 90 mM sodium acetate). This mixture was filtered through Qiagen Minelute spin columns to bind and wash DNA which was followed by two consecutive elutions with 50 μl Qiagen EB buffer.

**Whole-genome libraries and prescreening**. In total, 29 double-stranded, blunt-end Illumina sequencing libraries were prepared following Meyer and Kircher[52] and sequenced these on Illumina HiSeq 4000 platform at low coverage (median c.13 million reads per library) (Supplementary Data 2). Libraries from five individuals (TEP03, TEP62, TEP83, ULU23, ULU31) contained >1% endogenous sheep DNA, while other libraries had negligible proportions.

**Hybridization capture**. In order to increase coverage, we enriched chosen libraries for 20,000 SNPs (see below) via in-solution capture hybridization using custom designed[53] MyBaits probes (Arbor Biosciences Inc.) (see Supplementary Data 3 for the SNP list). We used four libraries with >1% endogenous sheep DNA (TEP03, TEP62, TEP83, ULU031) and one library with <1% endogenous sheep DNA (ULU26) (Supplementary Data 2). The biotinylated RNA capture probes were produced by Arbor Biosciences Inc. and capture experiments were implemented following the manufacturer's instructions. Specifically, hybridization reactions were incubated at 55 °C for 24 h and the captured libraries were amplified using Herculase II Fusion Polymerase (Agilent Technologies) for 15–19 cycles. Enriched libraries were purified using AMPure XP beads and then quantified by Agilent 2100 Bioanalyzer. Purified libraries were pooled in equimolar concentrations to reach a final concentration of 10 nM and then were sequenced paired-end with a 2 × 150 setup using HiSeq X SBS chemistry. We note that ULU26 coverage was only 0.003× after the second round of sequencing and data from this sample was not included in population genetic analyses using nuclear data, but its mitochondrial DNA was used in haplogroup assignment.

**SNP selection and capture probe design**. For in-solution-hybridization capture, 20,000 SNPs were selected from Illumina's OvineSNP50 Genotyping Beadchip SNPs[12], giving priority to transversions and also including mitochondrial markers and SNPs associated with putatively positively selected regions. Four 60-nucleotide-long probes were designed per SNP, summing up to a total of 80 K probes for 20 K SNPs. Two of the probes, carrying either the reference or the alternative allele, were centered on the SNP, while the other two (carrying the reference sequence) were located at each side of the targeted SNP, following the design by Haak et al.[54]. The 20,000 SNPs included all 8850 transversions present in the Illumina OvineSNP50 Beadchip variant set, 1237 chromosome X transitions, three mtDNA transitions, and 521 putatively functional SNPs (based on Kijas et al.[12], Moradi et al.[55], Heaton et al.[56], Noce et al.[57]). The remaining 9389 SNPs were randomly selected among OvineSNP50 transitions. The designed probes were produced by Arbor Biosciences Inc.

**Data preprocessing and mapping**. ANS data were processed following Günther et al.[58] and Kılınç et al.[34]. BAM files from shotgun and capture libraries from the same individual were combined, the residual adapter sequences in FASTQ files were removed and paired-end sequencing reads were merged using Adapter-Removal (v.2.3.1)[59], with an overlap of at least 11 bp between the pairs. The merged reads were mapped to the sheep reference genome oviAri3 (Oar_v3.1), which includes 26 autosomal sequences, as well as chromosome X and the mitochondrial genome. For mapping, we used the BWA (v.0.7.15)[60] "aln" algorithm. All libraries from the same individual were merged using SAMtools (v.1.9)[61] "merge" command, the PCR duplicates were removed using FilterUniqueSAMCons.py[52] and reads shorter than 35 base pairs were also removed. The number of mismatches was proportioned to read length and then the reads with a ratio of >10% were removed. Ancient individuals' BAM files were trimmed from both ends by 10 bp using bamUtil (v.1.0.14)[62] "trimBAM" command to avoid postmortem damage at read ends being interpreted as true variants.

**Genotyping**. We used the The Illumina OvineSNP50 Beadchip SNP panel for genotype calling. This list is based on oviAri1.v1 reference genome coordinates and were mapped to oviAri3 coordinates using the UCSC Genome Browser[63] "liftover" tool. Of the 49,034 SNPs in the SNP chip dataset, 40,225 could be mapped to the oviAri3 genome version and were used in subsequent analyses. For genotype calling, the SAMtools (v.1.9)[61] "mpileup" command was run on BAM files. The data were pseudohaploidised by randomly selecting a single read to represent the genotype[58]. The data belonging to the three OBI libraries were processed in the same manner as ANS data.

**PMD filtering**. PMD patterns at the first 60 positions at 5' and 3' ends of the reads that are specific to ancient DNA and not expected in modern DNA data were quantified by PMDtools (v.0.60)[64]. PMD profile graphs were generated using the PMDtools (v.0.60)[64] "–deamination" parameter and DNA molecules bearing the postmortem damage signature were selected using PMDtools (v.0.60)[64] with the "–threshold 3" parameter.

**Molecular sex determination**. The molecular sex of the four individuals was estimated by comparing the number of reads mapping to autosomes and chromosome X, after duplicate removal and quality filtering. The number of mapped reads was normalized by the length of the chromosome. Libraries of XY male individuals are expected to have reads mapping to chromosome X at an intensity of

50% compared to those mapping to autosomes (relative to the chromosomes sizes), unlike XX females, which are expected to have 1:1 intensity ratios. We calculated expected values of X intensity and confidence intervals under the null hypothesis of an individual being XX, using loess regression between chromosome size and the number of reads mapped. Graphs were plotted with the R (v.3.5)[65] library ggplot2[66], using the "stat_smooth" function to construct a regression curve with the loess method, and a span value of 0.80 (fraction of points used to fit the curve) to control the amount of smoothing. The regression models were constructed for each individual separately. The X chromosome data were not included in the regression model. Confidence intervals (95%) were constructed using a t-based approximation.

**Modern sheep genotypes.** For comparative analyses, we chose a subset of eighteen modern sheep breeds that would be representative and relevant to our study, out of 74 worldwide breeds (SheepHapMap) included the Kijas et al.[12,31] SNP chip dataset, excluding breeds known to have undergone strong bottlenecks (e.g. Soay) or recent admixture (e.g. Creole). This modern genotype data and our ancient genotype dataset were merged using the PLINK (v.1.9)[67] "mergelist" command, to produce a dataset with eighteen modern populations and four ANS samples.

**Principal component analysis.** Principal component analyses (PCA) were conducted using EIGENSOFT (v.7.2.0)[68] "smartpca" command with "lsqproject:YES" parameter Components of individuals from eighteen modern populations from the SheepHapMap project dataset were first calculated, and the four ANS individuals were projected onto the first three components. Visualization of the PCA was done by the R (v.3.5)[65] library ggplot2[66].

**D-statistics.** D-statistics were calculated using the AdmixTools (v.5.1)[33] "qpDstat" command. Modern, ancient, and outgroup datasets were merged using the "mergelist" command; extra chromosomes of sheep not recognized by PLINK (v.1.9)[67] were excluded by "exclude" command. The 'Plink' file format was converted into "eigenstrat" file format by the AdmixTools (v.5.1)[33] "convertf" command, before the D-tests were performed. The P values were calculated from Z scores based on the block jackknife procedure[33]. In order to control for the false discovery rate, multiple testing corrections using the Benjamini–Hochberg method[69] for each batch of tests were performed using the R (v.3.5)[65] "p.adjust" function.

**$f_3$-statistics.** Outgroup $f_3$-statistics were calculated using the AdmixTools (v.5.1)[33] "qp3Pop" command. For calculating outgroup $f_3$-statistics, modern and ancient datasets were merged and converted into eigenstrat file format using the same procedure applied in D-statistics. The significance of the correlation between the resulting $f_3$ values for each pair of ancient individuals and between $f_3(Argali; OBI, Modern)$ and $f_3(Argali; ANS, Modern)$ outgroup test were calculated using the Spearman correlation test by R (v.3.5)[65] "cor.test" function, the significance of the difference between values and European-ANS comparisons vs. non-European-ANS comparisons was calculated using the Mann–Whitney U test by R (v.3.5)[65] "wilcox.test" function.

**ADMIXTURE analysis.** Unsupervised clustering analysis was performed using ADMIXTURE (v.1.3.0)[35]. The 18 modern sheep-breed datasets were first filtered for linkage disequilibrium using PLINK (v.1.9)[67] with parameters "–indep-pairwise 50 5 0.05". All SNPs filtered ($n = 2265$, 19.5%) from the modern dataset were also excluded from ancient individuals using PLINK (v.1.9)[67]. Cluster analysis for modern sheep breeds was carried out using between $K = 2$ and $K = 12$ components. For every K value, we generated ten replicate runs, repeating the analysis using different random seed numbers. Finally, ADMIXTURE (v.1.3.0)[35] "projection" functions were used with random different seeds and again ten runs were performed following, to calculate ANS individuals' ancestral components based on those calculated from modern breeds. Pong (v.1.4.9)[70] was used for visualization. For each modern sheep breed, eight individuals were randomly chosen to improve visualization.

**SNPs at putatively selected loci.** Kijas et al.[12] had identified 31 putatively selected loci using an $F_{ST}$ scan across global sheep breeds, comprising the most extremely differentiated 0.1% of genome-wide markers. We compared these ancient sheep genotypes with those of 18 populations of modern domestic sheep breeds (used in the main analyses), as well as with two outgroups, Argali and Vignei, to identify the ancestral state. Data for both the outgroups and the modern sheep were retrieved from the Kijas et al.[12,31] dataset. We also considered studying putatively selected loci identified by Li et al.[16] but since none of the SNPs at these loci were present in our dataset, our analysis was limited to the aforementioned 31 SNPs.

**PCR amplification and sequencing of mtDNA.** Amplification of two sets of aDNA extracts from 180 samples for 144 bp mtDNA control region (CR) was performed in a volume of 20 μL containing, 1× Taq buffer, 2 mM MgCl₂, 0.25 mM dNTPs, 15 pmol of each primer, 2 ng BSA, and three unit of Applied Biosystems Amplitaq Gold 360 (5 u/μL) DNA polymerase (Fisher Scientific Inc.) Thermo-cycling conditions were as follows: initial denaturation at 94 °C for 10 min, followed by 60 cycles of denaturation at 94 °C for 30 s, annealing at 53 °C for 45 s,

extension at 72 °C for 45 s, and a final extension at 72 °C for 5 min. PCR amplicons were run on 2% agarose gel to observe successful amplification and assess contaminations in negative controls. The 144-bp long fragment of mtDNA corresponding to the positions 15,391–15,534 on the reference mitogenome (AF010406[71]) was sequenced from ancient samples by using primer pairs from Cai et al.[39] Sanger sequencing of the PCR products were carried out by Refgen Inc.

**mtDNA haplogroup assignment.** Our general strategy for haplogroup assignment was as follows: (i) In Sanger sequencing, we aimed to produce at least two 144 bp sequences per sample; (ii) for each sample, we compared the sequences to determine any discrepancies; (iii) in case of a discrepancy, we amplified and sequenced the third round, and used the consensus genotype; (iv) in samples where sufficient mitogenome data (>1500 bp coverage) was available, we also used this data to identify haplogroups (v) in two cases (TEP03 and ULU31) where there was the discrepancy between the Sanger-based assignment and mitogenome-based assignment, which appears to arise due to convergent transitions (homoplasy) at diagnostic positions, we reported the mitogenome-based result (see Supplementary Note 1). Samples were assigned to mtDNA haplogroups (A to E) according to the identity of nucleotides on haplogroup-determining positions with respect to the reference AF010406[71,72] sequence (Supplementary Table 7). The performance of the 144-bp fragment was also checked for accurate identification of haplogroups (see Supplementary Note 1 for details). Sample ULU26 in Supplementary Table 1 was also utilized for the haplogroup assignments. Mitogenome data was called from whole-genome sequencing by SAMtools (v.1.9)[61] "mpileup" command and BCFtools (v.1.9)[61], "call" command. For haplogroup assignment based on mitogenome data, VCF format was converted to FASTA format using custom python 3 code[53]. The best nucleotide substitution model was determined according to the Bayesian information criterion (BIC), as TN93 + G[73] using MEGA-X[74]. We again used MEGA-X[74] to construct a neighbor-joining (NJ) tree with 1000 bootstraps.

**Mitochondrial haplogroup frequency tests.** The significance of pairwise comparisons of haplogroup frequency differences between ancient sheep from central Anatolia and west Anatolia with modern domestic sheep from central Anatolia (Akkaraman), west Anatolia (Sakız and Kıvırcık) as well as Asia (China, Mongolia, India) and Europe (eastern, central and western) were tested using Fisher's exact by R (v.3.5)[65] "fisher.test" function and the effect size was estimated by Cohen's w[75] by R (v.3.5)[65] "cohenW" function.

**Statistics and reproducibility.** We performed all statistical analyses in (v.3.5)[65] via RStudio (v.1.3)[76]. All statistical tests were two-sided and nonparametric unless otherwise stated, p-values less than 0.05 were significant. Sample sizes and the number of replicates are reported where applicable. All laboratory procedures and parameters of statistical analyses are reported in the "Methods" section.

**Reporting summary.** Further information on research design is available in the Nature Research Reporting Summary linked to this article.

## Data availability
All FASTQ files were submitted to the European Nucleotide Archive (ENA) with reference number PRJEB36540[77]. All ancient mitochondrial DNA sequences produced in this study have been deposited in the NCBI GenBank database (accession number MT321187- MT321260)[78]. All of the ancient bone specimens analyzed in this study have been deposited in the ancient DNA laboratory at the Middle East Technical University (see Supplementary Data 1 for specimen numbers). These samples may not be shared and exported abroad due to the regulations implemented by the Republic of Turkey Ministry of Culture and Tourism. Modern DNA data from Kijas et al.[12] are accessible via the International Sheep Genomics Consortium (ISAG) website[31] (https://www.sheephapmap.org/download.php), Australian Government's data portal (https://data.gov.au/dataset/ds-dap-csiro%3A6494/details?q=) and Australia's National Science Agency CSRIO's website (https://data.csiro.au/collections/collection/CIcsiro:6494). Ancient DNA data from Taylor et al.[20] are available European Nucleotide Archive (ENA) with reference number PRJEB41594[32].

## Code availability
The code for probe design and the code for converting mitogenome VCF to FASTA are available at Zenodo[53].

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

## Acknowledgements

We are grateful to the METU CompEvo group, Daniel Bradley, Arpat Özgül, Cosimo Posth, and three anonymous reviewers for support, helpful suggestions, and/or comments. This work was supported by TÜBİTAK 1001 (Project No: 111T464 and 114Z356) and ERC Consolidator grant "NEOGENE" (Project No.: 772390).

## Author contributions

İ.T., M.Somel, F.Ö., and A. Götherström designed and supervised the study; D.B., Y.G.Ç., E.B., Ö.Ç., F.G., L.M., R.Ö., E.P., and J.A.L. equally contributed archaeological samples, archaeological information, discussion and writing up of results; C.Y.G., M.Şevketoğlu., and J.H. equally contributed archaeological samples and archaeological information; O.Ö., E.Yüncü., N.D.D., A.A., S.Y., V.K.L., M.K., M.F.A., and F.Ö. performed the wet lab experiments; E.Yurtman., O.Ö., E.Yüncü, D.Koptekin, D.Kaptan, and G.A. performed data analysis with contributions from M.Ö., K.B.V., G.M.K., A.Ghalichi., S.C.A., R.Y., E.Sağlıcan., N.E.A, C.C.B., T.G., P.M.M., A.Götherström, M.Somel, and İ.T.; and M.Somel, İ.T., F.Ö., E.Yurtman., E.Yüncü., O.Ö., D.Koptekin., Y.S.E., Ç.A., and E.Sürer wrote the manuscript with input from all authors.

## Competing interests

The authors declare no competing interests.
