## [Peer Review File · Communications Biology]

Reviewers' comments:

Reviewer #1 (Remarks to the Author):

Yurtman et al. present genetic analyses of sheep fossil from Central and West Anatolia and use these data to discuss sheep domestication history. I found the manuscript well-written, if a both lengthy for the amount of data they retrieved, but this may also be personal taste.

However, there are also a few issues that the author require to address in my opinion.

Generally, the entire mtDNA part is comparatively weak. First, 144 bp is a really short fragment, and it is known that such short fragments often conceal information (maybe best shown in Keis et al. 2012 *JBiogeography*). Second, I am surprised that despite at least two amplifications per individual (an important piece of information buried in the supplements), some sequences were removed because of post mortem damage. This should not happen if appropriate procedures are in place (i.e. third amplification in case of discrepancy). The author should elaborate more on their procedures. Next, the statistical treatment of the mtDNA data is a bit too creative for my taste. Calculating a haplotype diversity from five samples is not something I would encourage, and the result of this analyses with the confidence interval exceeding 1 (Supplementary Table 5) shows that this is indeed not recommended. Also I would like to point out that the results of the authors themselves show that there is no significant difference in haplotype frequencies in CA between the time periods >7,500 and 7,500-7,000 years. Therefore, the claim that the diversity from >7,500 years vanished in the following period is statistical not supported. I understand that, given the fact that all three samples from the latter period show the same haplotype, it is tempting to make this statement, but I nevertheless ask the authors to remove this claim. Last, that a haplotype based on 144 bp is present 6,000 years before domestication is actually not surprising. Overall, I suggest that the authors dampen down their conclusions based on the mtDNA results.

Otherwise I only have one other case of over-interpretation in lines 190-191: „The excess (albeit non-significant) of females is consistent with young male slaughter patterns.“ I suggest that the authors remove this statement – with four samples and two sexes, there are a total of only five different combinations and only one (2:2) would not count as excess under this definition.

Reviewer #2 (Remarks to the Author):

Yurtman and colleagues have generated archaeogenetic data from 74 Anatolian sheep, including four from the Neolithic for which they also generate genome-wide data. They show that there was a likely pre-domestication bottleneck and that Neolithic Anatolian sheep are closely related to modern sheep from north and central Europe, but, surprisingly, not to the geographically-closer eastern Mediterranean populations. These results add new genetic insights to the sheep domestication story, and are congruent with the archaeological record.

1a. For the majority of their samples (70/74), the authors generated a short fragment of the mitochondrial control region using PCR and Sanger sequencing. They extracted each specimen twice (line 350) and performed independent replication, which gives confidence in the data. However, there is no mention of negative controls (at both the extraction and amplification steps) to monitor for modern contamination. Please include these details and whether any sheep DNA was detected, as this is critical for evaluating the reliability of these results.

1b. The authors present their mitochondrial haplotype data in Figure 2 using a symbol and colour scheme. This does not say anything about the relatedness of the haplotypes to one another. I suggest that the authors instead use a phylogenetic network to present these data, perhaps colour-coded based on time bin.

1c. Related to the point above, there is no mention of which mitochondrial haplogroup is ancestral. Given that the samples span ~10,000 years, this short fragment is from the fast-evolving control region, and that a previous estimate put the tMRCA of modern sheep breeds at 800 generations (presumably less than 10,000 years; line 76), could it also be that haplogroup B is ancestral and

other haplogroups (e.g. A, C) had not yet evolved in the early periods? A discussion of whether this could be the case (or not) should be mentioned in the text.

1d. Asian and European sheep are described as having 'distinct proportions' of mitochondrial haplogroups A and B (lines 61-62). This is also eluded to on lines 293-294. It would be very helpful to add this information to Supplementary Table 2, so the reader can understand what these proportions are.

1e. Was the alpha threshold for the haplotype diversity tests (supplementary table 6, lines 391-399) corrected for multiple comparisons?

2. I consider the analyses of the nuclear single nucleotide polymorphism data set to have been well executed and have no major issues with these results. I particularly applaud the authors for the detailed evaluation of the authenticity of their data (damage patterns, and correcting for these) and thoroughly examining whether longer DNA fragments may have resulted from modern contamination.

2a. The authors consider their long read lengths to indicate unusual DNA preservation at their Anatolian sites (lines 185-187). Whilst that may be true, an alternative explanation is that short DNA fragments were lost during library preparation, thereby resulting in a longer-than-expected mean read length. To this end, it would be very helpful if the authors plotted their DNA fragment length distributions and highlighted the mean fragment length, as an alternative to supplementary figure 4.

2b. In the methods (SI line 403) and supplementary table S1, the authors state that libraries from five sheep specimens were target-enriched, yet the discussion only mentioned four (lines 155-158). Why was the remainder (ULU26) excluded?

2c. In supplementary table S7, there are low correlations when the <70 bp data set is used. I suspect this is the source of the 'low SNP numbers' (SI line 152). It would be helpful if the authors included a supplementary table with the number of SNPs in each length class for each sample after PMDtools filtering.

2d. In supplementary table S8, the linear model should not include the X-chromosome data (as evidenced by the confidence interval bulge). These data can then be plotted a posteriori.

3. To explain the difference between Neolithic Anatolian sheep and extant eastern Mediterranean populations, the authors assume a scenario of Asian introgression into the latter (lines 309, 322, 331). The authors should justify why Asia is the presumed source and not other regions, such as Africa. Could this also be explained by population replacement rather than admixture?

The authors analysed c.200 bones (line 88), but results are only presented for the 74 that were successful. Please add details for the specimens that did not amplify and for those that gave inconsistent mitochondrial results.

There are 74 samples mentioned in supplementary table S1, 78 in supplementary table S1, and 76 on line 374. Which numbers are correct?

The radiocarbon date information is in multiple places: ¹⁴C dates are in Supplementary Table S1, ¹⁴C date accessions are in the methods (lines 342-345), and calibrated dates are in the methods and Table 1. Please locate these in one place, preferably Table 1. How were these dates calibrated?

To enable replication, the authors should list their 20,000 SNPs (line 162) in a supplementary file, including genomic coordinates and whether they are transversions, mitochondrial, or domestication-related.

Figure 3 and supplementary figure 5a: please re-plot these on just the first two eigenvectors, as it is impossible to visualize a 3D plot in a 2D image. In supplementary figure 5a, consider making

the legend larger for readability.

Figure 4 is missing the data from the ancient individuals (ULU, TEP)... I assume these are similar to NSP and FIN?

Methods: there is a lot of overlap between the methods in the main text and in the supplement (e.g. lines 382-386 and 396-402 in the main text are highly similar to the supplementary text). This makes the text confusing, as some details are in either one set of text or the other. To aid the reader, I therefore recommend that the authors combine the two methods sections into one in the main text.

36 libraries were constructed (line 368) from 36 samples (SI line 401). However, only 29 are listed in supplementary table S8. Why the discrepancy?

State what read lengths and whether paired-end chemistry was used for both the HiSeq-4000 (SI line 402) and HiSeq-X (SI line 410) sequencing.

Check the Supplementary Figure and Supplementary Table calls - many are out of order and/or mis-assigned (e.g. lines 121, 255, SI lines 94, 463)

Minor:

line 26: define 'BCE'.

line 29: specify that these are nuclear SNP data.

line 38: state when the Neolithic transition took place.

line 75: are these 'local stocks' referring to within Europe or across the whole distribution of domesticated sheep?

line 75-76: is this tMRCA based on mitochondrial DNA?

line 76: approximately how far back in time are 800 sheep generations?

line 85: earlier in the intro, give the age range of the Neolithic.

lines 88-89: give exact number of bones analyzed and the age ranges of these within the 'early Holocene'.

lines 89-90: give the ages of the Epipaleolithic and Chalcolithic in Anatolia. It may be helpful to give these in the Figure 2 headers, and then refer to this figure.

line 95 and throughout: check order of supplementary tables.

line 126: state that these values are Shannon diversity indices.

line 127: change 'haplotypes' to 'sequences'.

line 143-144: what evidence is there to suggest that lineages were introduced from the region east of the Fertile Crescent?

line 156: correct '%2'.

line 166-167: state here that 10 bp were trimmed from either end of the reads.

lines 217-218, 227-228: repetitive definitions - only need to define once.

line 225: change 'demographic histories' to 'admixture histories', as other aspects of demography (e.g. population size) were not investigated.

line 247: change 'strains' to 'individuals'.

line 257: rephrase 'chose' to 'grouped with' or similar.

line 272: change 'we are witnessing' to the past tense.

line 358: specify which sample.

Figure 1 and supplementary figure 1a: enlarge the text on the figure, as it is currently difficult to read. If this makes the figure too busy, consider numbering the sites in red on the figure and defining these in the figure legend.

SI line 144: specify that these are 'captured libraries'.

SI lines 183, 184, 253, 279: change 'kms' to 'km'

SI line 193-194: use 'BCE' to be consistent with the main text.

SI line 205: define 'NISP'.

SI lines 226, 229, 236, 241, 260, 288, 313, 444: check for typos.

SI line 227: which sample?

SI lines 298-299: change to decimal point.

SI lines 366: change 'sandblasting' to 'abrasion'.
SI lines 378: what brand and company of Taq Polymerase?
SI lines 427-428: how were these reads removed?
SI line 489: change 'dataset were merged with' to 'datasets were merged'.

Supplementary table 1: what is 'depo' in the lab ID names?
Supplementary table 8: There are five libraries highlighted with an asterisk, but the legend states 'n=4'. State whether the 'number of reads mapping' is after duplicate removal and/or filtering for mismatches.
Supplementary figure 3: these lines do not refer to T, A, G, and C. Blue is C>T, and red is G>A.
Supplementary figure 9: are these outgroup- or admixture-f3 statistics?
Supplementary figure 10: on the x-axis, suggest annotating whether populations are European, Asian, or African.

Reviewer #3 (Remarks to the Author):

This paper reports a substantial number of mtDNA control region sequences from ancient sheep across Central and Western Anatolia, spanning from pre-Neolithic to Chalcolithic time periods. This data suggests i) a reduction in haplotype diversity within haplogroup B following c. 7,500 BCE, and (ii) an increase in haplogroup diversity from 7,000 BCE to 5,500 BCE, a trend which apparently continues to modern day breeds in the region. The paper also presents shotgun and capture-enrichment data for four sheep from the approx. 7,000-6,000 BCE era, which they use in concert with modern sheep breed SNP data to explore their relationship with later domestic populations. The key arguments made from this data are (i) Anatolian Neolithic Sheep (ANS) show more shared drift with European breeds than non-European, (ii) in particular North/Central European breeds, and that (iii) there is a degree of discontinuity between ANS and all modern sheep breeds (with technical caveats); these three points are linked to later a spread of Asian breeds prior to modern breed differentiation

This paper represents a substantial step forward in our understanding of the genetics of ancient sheep populations in late Epipaleolithic-Neolithic-early Chalcolithic Anatolia. To my knowledge this is the first ancient mtDNA data from Anatolian sheep during the crucial periods encompassing the onset and intensification of management. Given the importance of this region for the development and spread of sheep herding, the potential importance of this paper and resource is considerable.

The authors of this paper should be commended on the clarity of writing that characterized the majority of the text and supplementary material; it was well structured and flowed well, and would help its appeal to a broad audience as well as a more specialized one. In addition there has clearly been a substantial amount of work performed to process the number of archaeological material of this data set, and represents an important genetic resource for those studying the history of animal domestication i.e. zoo-archaeologists and archaeo-genetics. Finally I was pleased with the extent of work apparent in the supplementary texts, from methods to the zoo/archaeological context of the sites the samples are drawn from.

However the study has limitations which were not addressed satisfactorily in the text as it stands. The use of a novel targeted enrichment probe set to generate the bulk of ancient nuclear data, the reliance of a modern SNPchip reference set for many tests, and finally the inclusion of transitions in their working SNP set, are all of concern. Combined with the low (relative to other ancient studies) number of called SNPs in their four enriched samples, data set quality stands out as a major technical limitation of the study. Additionally, as presented the text and figures are not clear on details and decisions which are key to assessing the strength of the results, and which underlying the conclusions. Finally, several of the conclusions (eg line 329 "probably locally domesticated", line 32 "Asian contribution to south European breeds") are over-reaching given the quality of the data.

Although this paper does represent an important step in expanding our understanding of the sheep domestication history, the technical issues must be addressed before publication, and the strength of the conclusions amended to be more conservative. Specifically, a local domestication in Central

Anatolia being captured from a small number of mitochondria sequences from a limited number of sites, and the European-Asian Bronze Age admixture conclusion, are at best weakly supported from the data here, and lean substantially on the weight of previously published studies.

Please note that this review was performed without access to the outgroup f3 statistics quoted in Lines 227-229, which were not present in the Supplementary File 1 provided.

Major Comments

Figure 2 of the paper presents issues with interpreting the underlying mtDNA data. The decision to not differentiate between sites within the two major regions (Central and Western Anatolia) is problematic, as it does not clearly indicate the uneven distribution of sequences from the various sites and regions. This is crucial given that the paper is arguing that the data captures a transition from wild to managed sheep - if this signal was being driven by just one site, the figure as stands would not reflect this. I note that the figure text highlights this - "Note that different periods and haplogroups are not homogenously..." - but this is not sufficient for a main figure. Additionally, this text contains a major issue: "while 7500-7000 BCE by Canhasan III and Tepechik-Çiftlik". In Figure 2 itself, this period is represented by three HPG B sequences of the same haplotype. When Supplementary Table 2 is examined, the three B sequences are all in fact from Canhasan III (CH1, CH2, CH3). Although some sequences from Tepechik-Çiftlik do have (estimated and C14) ages which partially overlap the 7,000 BCE boundary, they appear to be grouped in the 7000-6000 group in Figure 2, and the associated text is somewhat misleading. The Figure should be restructured in order to express the archaeological origin of the ancient samples here. The text should specify how samples are placed in the different periods when their ages straddle the transition.

Furthermore, Figure 2 does not include mtDNA haplogroups identified by NGS, not PCR - specifically the HPG sequence from Tepechik-Çiftlik (TEP83; Table 1). All other samples from Table 1 were included in the PCR-based sequencing and are presented in Figure 2. Although the NGS data may be insufficient for accurate haplotyping, it appears to be sufficient for haplogroup assignment (see comment below on NGS mtDNA methodology). As TEP83 represents the only HPG A sequence from their data, it is of interest to many readers and bolsters the paper's argument for an increase in haplogroup diversity post 7000 BCE. The sample - at least the presence of HPG A in the PrePottery Neolithic - should be indicated in Figure 2.

The modern sequences in Figure 2 reflect the samples chosen for haplogroup comparison, rather than an accurate representation of haplogroup frequencies in modern Turkey (Supplementary Table 2). For example the central Turkish breed Akkaraman has 52% HPG B in Sup Table 2, but according to Figure 2 sheep from the region have $4/21 = \sim 19\%$ freq of HPG B. This is quite misleading, over-estimating the apparent change in haplogroup frequencies, and the apparent difference in changes in mtDNA diversity between Western and Central Anatolia. This subsequently affects Haplogroup diversity estimates (Sup Table 5). The authors should have downsampled the Central Anatolian 144bp sequences to reflect their actual present day frequencies and repeat the haplogroup diversity analysis.

High data quantity and quality is a luxury rare in ancient DNA, and inferences are often limited by this. This paper does present a substantive jump forward in the genomic data from ancient sheep populations, but SNP coverage is still low (3,294-10,484). Given this, it is essential that uncertainty from standard errors and number of snps in each test be presented for all tests, as they are not in the present text or supplementary materials/supplementary file 2. Additionally, the D statistic tests quoted lines 216-225 should be included in the supplementary files, particularly with the number of ABBA and BABA sites for each, as the lack of statistical significance could be a consequence of low SNP number.

Regarding the modern breed data, the text is often confusing about whether it is referring to geographic regional groupings or groupings from the original source (Kijas, 2012). For example, does "Eastern Mediterranean strains" refer to the Lecce breed? Is Sakiz an "Eastern Mediterranean" breed, while being grouped in "SW Asia" in Sup Table 7? In lines 248-253, are the "central", "northern", and "south" European breeds referring to the groupings in Sup Table 7? Similarly, lines 311 mention "Middle Eastern", "south" and "central/north European" sheep." The authors should be clear in the text and Sup Table about which grouping they are using, ideally delineating their terminology in Sup Table 7.

The authors have chosen to perform analyses using both transitions and transversions, while

aggressively trimming read ends by 10bp to reduce the ancient damage present. Although deamination-related damage is predominantly at the ends of fragments, it clearly persists in this data further than 10bp from read ends (Supplementary Figure 3). This could very plausibly confound key tests, such as the test suggesting a recent homogenization of modern sheep breeds (line 256, D(Argali, modern1, modern2, ANS). ANS may show persistent error and resulting conversions of the derived B allele to a non-B allele, inflating the relative number of derived alleles shared between modern1 and modern2 (this should not be an issue when ANS is used to define the derived allele e.g line 244-245). This may also confound the test arguing that ANS individuals are more similar to each other than with any modern breed (line 217, D(Argali, ANS1, ANS2, modern)- note that these tests were not provided in the supplementary files and ideally should be). Although in the text "technical issues" are discussed (lines 239-321), the inclusion of transitions in analyses present a glaring potential source of error. This may also confound the outgroup f3 statistical tests e.g. (line 227, f3(Argali; modern, ANS). This may exacerbate issues inherent to the study design: the nature of the reference genome (North European-derived breed) and targeted enrichment may introduce reference bias, making poorly covered samples (i.e. the ANS) more similar to the North European-like reference. Given that several conclusions are drawn from apparent ANS affinity with North/Central European breeds, this is concerning. Potential reference bias in alignment and enrichment is likely outside the scope of this study. At a minimum, key statistical tests should be repeated restricting to transversion only. This may reduce SNP numbers substantially; demonstrating correlation between test results with and without transitions may be sufficient in this case to argue against transitions-error driving results. Full information (D, f, Z, number of snps) for both snp sets (transitions+transversions, transversions-only) should be provided in the supplementary file.

Following both comments 4 and 6, key statistical tests are lacking from Supplementary File 1. The authors state (lines 227-229) that outgroup f3 statistics using Argali sheep as the outgroup were presented in this file, but these tests were not in the file provided by the authors. Given that these statistics underlie a central result and figure (Figure 4), these tests must be provided to readers. Additionally, being able to directly assess the standard errors associated with each test would be very useful for readers, given how similar the median values are for the European vs non European shared drift (albeit statistically distinct based on the Mann Whitney U-test, lines 241-243). Finally, why did the authors not employ the same statistical test for determining differences in the shared drift with North/Central European breeds vs South European?

The major comments above (and some minor below) lead me to conclude that the present study overstates their data power to support one of the key conclusions, line 31-33: "Our results indicate that Asian contribution to south European breeds in the post-Neolithic era, possibly during the Bronze Age." The key three pieces of evidence supporting this have issues. As discussed, the D statistic bias towards allele sharing between ANS and North/Central European breeds may be the product of reference bias. ADMIXTURE results (see below) do not confidently establish a difference in ancestry similarity between North, Central, and Southern European breeds, so it is selective to state that ANS show an ancestry profile most similar to N/C European breeds (lines 253-254). And finally, as the authors acknowledge (lines 258-259), the D test indicating greater allele sharing among modern breeds than with any modern breed and ANS is possibly a technical issue, which could be the persistence of deamination error in the data (see below). The D statistic tests using just modern breeds are not sufficient (lines 308-313), as we are not presented the number of tests which do not support the affinity between Middle Eastern and South European breeds, and authors do not raise a very plausible alternative hypothesis - that the two regions, being geographically close, experienced greater trade and gene flow. As such, unless the authors present transversion-only results which also support this conclusion, it is difficult to conclude that the data supports this major conclusion.

Following from the previous comment, the authors introduce some hypotheses that their data does support, even without the issues discussed. Specifically, the Bronze Age is not discussed in any detail outside of the introduction of wooly breeds to Europe in the Introduction (31-33). If the genomic data was from Europe and/or Bronze Age contexts, such inferences would be fair, but currently the data from Neolithic Anatolia is too weak to support this kind of speculation in the Abstract.

3D PCAs (Figure 3) are very difficult to interpret on a 2D plane. At the very least they should consider pairwise plots of PCs presented in the SI, if not replacing Figure 3 with a 2D equivalent. In addition, transition error could be driving the "relative central location" (line 214) of the ANS samples in the PCA. The PCA should be repeated using transversions-only and presented in the SI.

Note that for this and the previous comment, transversions-only may be too few sites for robust population analysis. The authors should consider determining the frequency at which ANS samples show the damage allele at transition sites in their data, compared to individual modern breeds or all modern breeds.

The authors report unusually long fragments for their ancient data. Although the issues raised above may confound the results, a few additional steps would aid in firmly establishing their authenticity. mtDNA haplotypes as determined from long vs short reads should be identical, once missingness and damage is accounted for; the authors check these against each other.

Additionally, the size distribution of aligned fragments should be presented for each sample; unimodality would support an ancient origin of the long reads.

The supplementary methods do not mention any control experiments e.g. for the DNA extraction or PCR amplification. Given that 60/74 of PCR mtDNA sequences from Figure 2 have the same haplotype, the same haplotype "most widespread today" (line 137), negative controls would be vital in ruling out modern contamination. Could the authors clarify if these were performed.

Similarly for Supplementary Table 8, there is no mention of negative control libraries included in the WGS and enrichment. Could the authors clarify if these were performed, preferably including the sequencing results of the control libraries e.g. proportion of reads aligning to sheep .

Line 101, "each likely from distinct individuals according to context (Supplementary Material and Methods)" - from the supplementary text, I could not see any indication how samples were determined to be "likely" distinct - could the others please clarify. Inclusion of material type (talus, molar, left petrosal etc) in Supplementary Table 1 could help this.

Minor Comments

The authors should touch on the identity of the wild progenitor of domestic sheep, and other relevant *Ovis* species, in the introduction, as it is relevant to later statistical tests.

Line 82, the authors mention "one of the possible domestication centres". The introduction should be more specific in what these potential alternatives centres are, and perhaps

Line 127 - "totally vanished" not appropriate language, change.

Line 146-147 - if the authors state that that the change in haplogroup composition "happens more subtly" in Western Anatolia, they should provide a reference to their results backing up the statement (Haplogroup diversity from Sup Table 5 etc).

Line 148-149, "analyses of maternal lineages lend support to a domestication event in central Anatolia" - the data is also consistent with domestic sheep being introduced to the region; the presence of the later-common HPG B type in pre 7,500 BCE Pınarbaşı is not sufficient to conclude that domestication occurred in Central Anatolia.

Line 141, "our sample size is too small to exclude the presence of non-B haplogroups" - four samples is certainly too few to draw this conclusion. The authors could easily explore this e.g. if the frequencies in Central Anatolia between 7500-7000 and 7000-6000 are same, the chances of sampling a non HPG B from three sequences, or the maximum frequency a haplogroup would have to be to sample at least one in three samples at say 0.95 probability. This would contextualize the speculation that follows.

Line 159 - according to Supplementary Table 2, the calibrated age of TEP03 also extends into the 8th millennium, not just TEP62. Please correct.

Line 190-191 - four samples is far too few to make any statement on sex bias. Three out of four sheep being female would be consistent with a bias towards males as well, given the level of statistical power. I recommend removing reference here to sex bias.

Line 224, "from different periods" - are the authors referring to the periods as defined in text i.e. 7500-7000 BCE and 7000-6000 BCE? As the four samples all have C14 ages which predominantly lie within the 7000-6000 BCE period.

Line 253-254 - "ADMIXTURE...indicated similarity between ancestry components in AND and north and central European breeds" - visually assessing all Ks, the ANS sheep also have similar profiles to south European breeds e.g. LEC; European breeds appear to have very similar profiles. Differences present are subtle at best and the authors do not present error bars on ancestry fractions in the ANS samples to assess ADMIXTURE results (low SNP number, transitions could produce high variation between runs). From this analysis alone, it is not possible to say that they show more similar profiles to any one European regional group, but could state a greater similarity in profiles to European breeds than non-European. The authors should amend the text to reflect this, and give some indication of the uncertainty of ADMIXTURE results on their data.

Line 270 and also line 26 - "loss of mitochondrial haplotype diversity" - the authors should be clear that this loss is specifically HPG B haplotype diversity. This applies to the entire paper; the authors often refer to "haplotype diversity" (e.g. line 99 in SI); if the haplotype diversity estimates are based solely on HPG B sequences, the text should indicate as such and not solely in Supplementary Table 5 (it is not clear from main text or SI methods).

Line 275-276 - acknowledging that the authors are careful just raise the possibility of a local domestication in Central Anatolia based on the presence of the later-ubiquitous lineage of HPG B, this is based from a single contextually-dated sample, not C14 dated. This should be indicated in the text.

Lines 298-299 - the strength of these conclusions may be affected by the technical issues discussed above, and the data may not have the power to support this.

Line 328-329, "central Anatolian wild sheep were probably locally domesticated" - again this inference is drawn from a single undated haplotype from Pınarbaşı. I do not think the current genetic data is strong enough to come to this conclusion. An introduction of domestic sheep to the region is also quite possible, as a single haplotype sequence is not sufficient to discount it. The discussed lines should be toned down or alternatives emphasized more, as the data is not sufficient to directly support these conclusions.

The authors do not specify how Illumina data was aligned to a mitochondrial reference; this should be added to the supplementary methods.

On line 358 the authors refer to inconsistent mtDNA genotypes for "one individual" possibly due to postmortem damage. Could the authors elaborate, and indicate which individual the inconsistent genotypes were generated from.

Supplementary Table 9 - in some cases it is not clear which base was taken as the ancestral base e.g. OAR3_141586525.1. Could the table indicate explicitly which base was taken as ancestral at each SNP.

Supplementary Figure 4 - could the authors also indicate the median read lengths, which would likely be a more informative statistic in this case.

Supplementary Figures 7 and 9 - please indicate on each comparison which correlation is significant. This is not clear at all for Sig Figure 9.

Supplementary File 2, although containing the tests it should (D statistics), does not have sufficient data for assessment. The number of SNP, ABBA sites, and BABA sites for each test should all be presented, ideally with the standard error (although these can be calculated from the D and Z score, it would be clearer to present all of this information together). Finally if the authors could clearly show which of the two sets of tests each individual test is included in (i.e. either lines 244-245 or line 249), this would add clarity for readers.

Reviewers' comments:

Reviewer #1 (Remarks to the Author):

Yurtman et al. present genetic analyses of sheep fossil from Central and West Anatolia and use these data to discuss sheep domestication history. I found the manuscript well-written, if a both lengthy for the amount of data they retrieved, but this may also be personal taste.

However, there are also a few issues that the author require to address in my opinion. Generally, the entire mtDNA part is comparatively weak. First, 144 bp is a really short fragment, and it is known that such short fragments often conceal information (maybe best shown in Keis et al. 2012 JBiogeography).

We concur with the Reviewer that short mtDNA loci cannot completely reflect the complex demographic history of a species. However, because ancient sheep bones from excavation sites could have been exposed to many DNA deteriorating factors and especially high heat during the cooking, we anticipated a low success rate in obtaining the full genomes of the ancient sheep. We thus focused on a control region (CR) fragment, a relatively informative region of the mitogenome¹, and had chosen a 144 bp long sub-region of the CR since it captures sites to identify the 5 haplogroups of domestic sheep^{2,3}.

We made the following changes to the manuscript to address the Reviewer's points:

- We tested the reliability of 144 bp fragment when diagnosing haplogroups by using 310 modern mitogenome sequences, the results of which are presented in the Supplementary Note 1. Briefly, the locus allows 95.5% accurate classification using modern-day data, but misassignment is a non-negligible possibility. Indeed, we identified two cases of D->B misassignment caused by an apparent convergent transition at a diagnostic site among two ANS individuals.
- We mention this result in the main text and note the possibility of misassignment.
- In the Supplementary Note 1 we further discuss the limitations of demographic inference from mtDNA data in general (referring to work such as Keis et al 2013⁴; Larson and Burger, 2013⁵).
- In light of these points, we decided to limit the focus on mtDNA results. We thus reordered the main text, such that we present the mtDNA results as auxiliary to those from nuclear DNA, i.e. the affinity of Anatolian Neolithic sheep to present-day European sheep.

Second, I am surprised that despite at least two amplifications per individual (an important piece of information buried in the supplements), some sequences were removed because of post mortem damage. This should not happen if appropriate procedures are in place (i.e. third amplification in case of discrepancy). The author should elaborate more on their procedures.

We thank the Reviewer for this point. We now explain our approach in detail in Materials and Methods at lines 598-618: (i) In Sanger sequencing, we aimed to produce at least two 144 bp sequences per sample; (ii) for each sample, we compared the sequences to determine any discrepancies; (iii) in case of discrepancy we amplified and sequenced a third round, and used the consensus genotype; (iv) in samples where sufficient mitogenome data (>1500 bp coverage) was available, we also used this data to identify haplogroups, (v) in two cases (TEP03 and ULU31) where there was discrepancy between the Sanger-based assignment and mitogenome-based assignment, which appears to arise due to convergent transitions (homoplasy) at diagnostic positions, we reported the latter result.

In the current version we did not remove any samples. Single copy amplifications of the 144 bp fragment are also presented in Figure 4 (**Figure R1** below) (hatched boxes).

We would also like to clarify the reason behind our earlier removal of haplogroup A sequences. We initially observed that TEP83, which has both Sanger and NGS data, was assigned haplogroup A with Sanger data, but assigned haplogroup B with NGS data. We hypothesized that recurrent postmortem damage (PMD) at diagnostic positions may have caused a B->A misassignment (as HPG B carries T where HPG A carries C). We reasoned that, if this happens, we could not trust other haplogroup A sequences as well (given the predominance of B in the ancient sample), so we removed all cases of HPG A, and reported this in the manuscript.

However, we later realized that the real issue was an error in our code (related to the Samtools parameters chosen when analysing the NGS data). Correcting this, the inconsistency disappeared – the haplogroup of TEP83 was identified as A also with the NGS data. We further reasoned that the possibility of PMD-induced transitions at the same position in >1 fragment is extremely low, as the Reviewer has rightly alluded to.

ARCHAEOLOGICAL SITES		PERIODS				
		Epipaleolithic/Pre-pottery Neolithic	Early Neolithic	Middle Neolithic	Late Neolithic	Present-day
		>7500 BCE	7500-7000 BCE	7000-6000 BCE	6000-5500 BCE	
CENTRAL ANATOLIA	Pınarbaşı					Akkaraman
	Boncuklu Höyük					
	Canhasan III					
	Tepecik-Çiftlik Höyük					
WEST ANATOLIA	Ulucak Höyük					Sakız
	Barcın Höyük					Kıvırcık

Figure R1. Change in mitochondrial lineages from Epipaleolithic to present-day in Anatolia. Distribution of different haplogroups (HPG) (indicated by colors) of the studied sheep individuals. Color coding is as follows: HPG A (red), HPG B (blue), HPG C (yellow), HPG D (green) and HPG E (orange). The hatched pattern indicates individuals supported by a single sequence. Note that different periods and haplogroups are not homogeneously represented across different sites from a single region: e.g., >7500 BCE in central Anatolia is represented by Pınarbaşı and Boncuklu Höyük, while 7500-7000 BCE by Canhasan III. See Supplementary Table 1 for full details. For the samples with ages straddling two periods, we used their mean age to place them within an archaeological period. Haplogroup compositions of present-day breeds of Turkey were represented by Akkaraman (central Anatolia) and by Sakız and Kıvırcık (west Anatolia) (see Supplementary Table 6). Haplotypes of ancient individuals are represented in Supplementary Figure 16.

Next, the statistical treatment of the mtDNA data is a bit too creative for my taste. Calculating a haplotype diversity from five samples is not something I would encourage, and the result of this analyses with the confidence interval exceeding 1 (Supplementary Table 5) shows that this is indeed not recommended.

We thank the Reviewer for raising this point. We removed haplotype diversity analyses completely from the manuscript. We now present mtDNA haplotypes in Supplementary Figure 16 and discussed them at Supplementary Note 1 at lines 268-350.

Also I would like to point out that the results of the authors themselves show that there is no significant difference in haplotype frequencies in CA between the time periods >7,500 and 7,500-7,000 years. Therefore, the claim that the diversity from >7,500 years vanished in the following period is statistical not supported. I understand that, given the fact that all three samples from the latter period show the same haplotype, it is tempting to make this statement, but I nevertheless ask the authors to remove this claim.

We agree with the Reviewer that this is an overstatement, and we removed haplotype-based results from the main text. We only briefly elaborate on possible differences in haplotype composition before and after 7500 BCE in the light of non-genetic data such as zooarchaeological observations and isotope analysis in Supplementary Note 2, with strong emphasis that our sample size does not allow any clear conclusions.

Last, that a haplotype based on 144 bp is present 6,000 years before domestication is actually not surprising. Overall, I suggest that the authors dampen down their conclusions based on the mtDNA results.

We agree with the Reviewer's point. We removed the haplotype diversity analyses from the manuscript. We now discuss our mtDNA results with utmost caution in lines 278-339 of the main text and lines 268-350 of the supplementary text.

Otherwise I only have one other case of over-interpretation in lines 190-191: „The excess (albeit non-significant) of females is consistent with young male slaughter patterns.“ I suggest that the authors remove this statement – with four samples and two sexes, there are a total of only five different combinations and only one (2:2) would not count as excess under this definition.

We agree, and now removed this point from the text.

Reviewer #2 (Remarks to the Author):

Yurtman and colleagues have generated archaeogenetic data from 74 Anatolian sheep, including four from the Neolithic for which they also generate genome-wide data. They show that there was a likely pre-domestication bottleneck and that Neolithic Anatolian sheep are closely related to modern sheep from north and central Europe, but, surprisingly, not to the geographically-closer eastern Mediterranean populations. These results add new genetic insights to the sheep domestication story, and are congruent with the archaeological record.

1a. For the majority of their samples (70/74), the authors generated a short fragment of the mitochondrial control region using PCR and Sanger sequencing. They extracted each specimen twice (line 350) and performed independent replication, which gives confidence in the data. However, there is no mention of negative controls (at both the extraction and amplification steps) to monitor for modern contamination. Please include these details and whether any sheep DNA was detected, as this is critical for evaluating the reliability of these results.

We apologise for this omission and thank the Reviewer for pointing it out. Negative controls for both DNA extraction and PCR amplifications were included throughout the experiments. This information is now added to the main text (subsection: “Procedures for contamination prevention and detection” at lines 428-445).

1b. The authors present their mitochondrial haplotype data in Figure 2 using a symbol and colour scheme. This does not say anything about the relatedness of the haplotypes to one another. I suggest that the authors instead use a phylogenetic network to present these data, perhaps colour-coded based on time bin.

We thank the Reviewer for her/his suggestion about the representation of the haplotypes. Now we present the relatedness of haplotypes with a phylogenetic network in Supplementary Figure 16b. We note that we removed haplotype results from the main text following suggestions by the reviewers that the data was too thin.

1c. Related to the point above, there is no mention of which mitochondrial haplogroup is ancestral. Given that the samples span ~10,000 years, this short fragment is from the fast-evolving control region, and that a previous estimate put the tMRCA of modern sheep breeds at 800 generations (presumably less than 10,000 years; line 76), could it also be that haplogroup B is ancestral and other haplogroups (e.g. A, C) had not yet evolved in the early periods? A discussion of whether this could be the case (or not) should be mentioned in the text.

We now explain this point in the main text at lines 288-289. Briefly, neither haplogroup A nor B appears ancestral within sheep mitochondrial DNA diversity. Indeed, Meadows et al.⁶ estimates that the tMRCA of the 5 haplogroups dates back to 900 K years ago, while HPG A and HPG B diverged only about 60 K years ago.

1d. Asian and European sheep are described as having 'distinct proportions' of mitochondrial haplogroups A and B (lines 61-62). This is also eluded to on lines 293-294. It would be very helpful to add this information to Supplementary Table 2, so the reader can understand what these proportions are.

We thank the Reviewer for this helpful suggestion. In Supplementary Table 6 (**Table R1** below), we now added the frequencies of haplogroups A-E for a number of sheep breeds reported by various sources, including the Mongolian Plateau, the Indian subcontinent, China, as well as eastern, central and western Europe.

Table R1. Mitochondrial haplogroup (HPG) frequencies of three present-day sheep breeds from Anatolia and different regions of Eurasia. References are shown in the “Source” column. The frequency of the most common haplogroup in each sample is shown in bold.

Region (breed) (sample size)	HPG A (%)	HPG B (%)	HPG C (%)	HPG D (%)	HPG E (%)	Source
China (n=463)	56	35	9	0	0	Lv et al. ⁷
Mongolian Plateau (n=245)	56	28	16	0	0	Luo et al. ⁸ Lv et al. ⁷
Indian Subcontinent (n=500)	77	20	3	0	0	Singh et al. ⁹ Gorkhali et al. ¹⁰ Lv et al. ⁷
Central Anatolia (Akkaraman) (n=50)	26	52	14	2	6	Demirci et al. ³
West Anatolia (Sakız) (n=49)	4	88	8	0	0	Demirci et al. ³
Northwest Anatolia (Kıvrıcık) (n=45)	0	96	4	0	0	Demirci et al. ³
East Europe (n=126)	8	92	0	0	0	Tapio et al. ¹¹ Činkulov et al. ¹² Lv et al. ⁷
Central Europe (n=19)	21	79	0	0	0	Tapio et al. ¹¹
West Europe (n=177)	4	94	2	0	0	Pereira et al. ¹³ Tapio et al. ¹¹

1e. Was the alpha threshold for the haplotype diversity tests (supplementary table 6, lines 391-399) corrected for multiple comparisons?

We now removed the haplotype diversity analyses from the manuscript.

2. I consider the analyses of the nuclear single nucleotide polymorphism data set to have been well executed and have no major issues with these results. I particularly applaud the authors for the detailed evaluation of the authenticity of their data (damage patterns, and correcting for these) and thoroughly examining whether longer DNA fragments may have resulted from modern contamination.

We thank the Reviewer for appreciating our work.

2a. The authors consider their long read lengths to indicate unusual DNA preservation at their Anatolian sites (lines 185-187). Whilst that may be true, an alternative explanation is that short DNA fragments were lost during library preparation, thereby resulting in a longer-than-expected mean read length. To this end, it would be very helpful if the authors plotted their DNA fragment length distributions and highlighted the mean fragment length, as an alternative to supplementary figure 4.

We thank the Reviewer for this suggestion. We added this data as Supplementary Figure 4a (**Figure R2a** below). Although it is not impossible that short fragments were lost specifically in the Tepecik samples during library preparation, because we do not observe long /fragments in the ULU31 data, which was prepared in the same batch as the Tepecik samples, we find this an unlikely scenario.

We also note that this “long fragment” pattern could be observed in at least 2 other Tepecik Neolithic sheep libraries (labelled as Tep001 and Tep006 in **Figure R2b**) subsequently processed and sequenced as part of a separate study and analysed by different researchers (Kaptan et al., unpublished). Again, these sheep genomes also showed the same population genetic characteristics as other Anatolian Neolithic sheep.

We also see unusual DNA preservation at this site, consistent with our earlier observations on Neolithic human material from Tepecik-Çiftlik as mentioned in the main text. Overall, these results support our hypothesis that the long reads are caused by exceptional preservation.

Figure R2. Frequency distributions of read lengths. The histograms represent lengths of reads that map to the sheep genome obtained from direct shotgun sequencing (blue) and sequencing after hybridisation capture (red) of libraries of ANS individuals. The spike at 150 bp is caused by the rare cases of fragments > 300 bp long. For these fragments, the forward and backward reads do not merge (as sequencing is run for 150 cycles) and the fragments are amassed at 150 bp, creating the spike.

2b. In the methods (SI line 403) and supplementary table S1, the authors state that libraries from five sheep specimens were target-enriched, yet the discussion only mentioned four (lines 155-158). Why was the remainder (ULU26) excluded?

We thank the Reviewer for pointing out this confusion. We had included a fifth individual, ULU26, in target enrichment. But the genome coverage went only up to 0.003X, which would not allow reliable population genetic analyses to be conducted. However, we utilized mtDNA fragments from this individual obtained from NGS sequencing. We now added this information at lines 467-469 of the Materials and Methods “Hybridization capture”, and at lines 611-612 of the “mtDNA haplogroup assignment” section in the main text.

2c. In supplementary table S7, there are low correlations when the <70 bp data set is used. I suspect this is the source of the 'low SNP numbers' (SI line 152). It would be helpful if the authors included a supplementary table with the number of SNPs in each length class for each sample after PMDtools filtering.

Thank you for the suggestion. This information is now added as Supplementary Table 4 (**Table R2** below).

Table R2. The number of SNPs in each length class for each ancient sample after PMDtools¹⁴ filtering.

Sample ID	No filter	Only transversion	<70 bp	>100 bp	PMD filtered
TEP03	15794	5492	3080	14716	11207
TEP62	20281	5843	5347	17983	13823
TEP83	6212	2349	1032	3767	3482
ULU31	8533	3437	6716	988	5172

2d. In supplementary table S8, the linear model should not include the X-chromosome data (as evidenced by the confidence interval bulge). These data can then be plotted a posteriori.

We thank the Reviewer for raising this point. In fact, we did construct our *loess* regression models only using autosomal data. We now explicitly mention this at lines 526-531 of the main text, as well as in the legend of Supplementary Figure 8 (and **Figure R3** below). The reason for the bulge is the lack of autosomes with certain size range, and chrX happening to fall within that range. We note the presence of the bulge in both TEP03 and TEP83 - one female, the other male.

Figure R3. Molecular sex estimation of Anatolian Neolithic sheep individuals used in genomic analyses. Sex was estimated based on comparison of number of reads mapping to sheep chromosome X and autosomes. For each ancient individual, for all chromosomes, the number of reads (y-axis) were plotted against the size of the chromosome (x-axis). Black dots represent autosomes while chromosome X is depicted as a red dot. In each plot, 95% confidence intervals were constructed using a t-based approximation by ‘loess’ function of R (v.3.5)¹⁵. Note that the regression model was constructed only using autosomal data, and chromosome X only used for prediction.

3. To explain the difference between Neolithic Anatolian sheep and extant eastern Mediterranean populations, the authors assume a scenario of Asian introgression into the latter (lines 309, 322, 331). The authors should justify why Asia is the presumed source and not other regions, such as Africa.

Could this also be explained by population replacement rather than admixture?

We agree with the Reviewer that Africa could be a possible source for gene flow into the Mediterranean. Indeed, Muigai and Hanotte¹⁶ suggest that thin tail sheep was first introduced to northeast Africa around 5500-5000 BCE, and fat tail sheep arrived there presumably after 1000 BCE. We therefore updated the main text at line 405, where we mention Africa as a possible source of introgression, alongside Asia.

We are currently not able to test full replacement vs. admixture. Therefore we replaced “admixture” with “gene flow”, to avoid implications regarding the extent of genetic change, at lines 395, 401, and 404.

The authors analysed c.200 bones (line 88), but results are only presented for the 74 that were successful. Please add details for the specimens that did not amplify and for those that gave inconsistent mitochondrial results.

We now added information about all 180 specimens that we studied, including those that did not amplify, as well as those with inconsistent haplogroup assignment in Supplementary Table 1. We also now present our strategy used in haplotype assignment and provide information on inconsistencies in Materials and Methods lines 598-618: (i) In Sanger sequencing, we aimed to produce at least two 144 bp sequences per sample; (ii) for each sample, we compared the sequences to determine any discrepancies; (iii) in case of discrepancy we amplified and sequenced a third round, and used the consensus genotype; (iv) in samples where sufficient mitogenome data (>1500 bp coverage) was available, we also used this data to identify haplogroups, (v) in two cases (TEP03 and ULU31) where there was discrepancy between the Sanger-based assignment and mitogenome-based assignment, which appears to arise due to convergent transitions (homoplasmy) at diagnostic positions, we reported the latter result. In Supplementary Note 1 lines 305-350 we provide more detailed information about these discrepancies and explain why we believe this was a genuine result caused by homoplasmy, and not an artifact.

We would also like to clarify the reason behind our earlier removal of haplogroup A sequences. We initially observed that TEP83, which has both Sanger and NGS data, was assigned haplogroup A with Sanger data, but assigned haplogroup B with NGS data. We hypothesized that recurrent postmortem damage (PMD) at diagnostic positions could have caused a B->A misassignment (as HPG B carries T where HPG A carries C). We reasoned that we could not trust other haplogroup A sequences as well (given the predominance of B in the ancient sample), so we removed all cases of HPG A, and reported this in the manuscript.

However, we later realized that the real issue was an error in our code (related to the Samtools parameters chosen when analysing the NGS data). Correcting this, the inconsistency disappeared – the haplogroup of TEP83 was identified as A also with the NGS data. We further reasoned that the possibility of PMD-induced transitions at the same position in >1 fragment is extremely low. Accordingly, we have included all samples which produced amplification in this version.

There are 74 samples mentioned in supplementary table S1, 78 in supplementary table S1, and 76 on line 374. Which numbers are correct?

We thank the Reviewer for noticing this mistake. The correct number of analysed samples is 79. This is corrected both on the main text at lines 302, 324 and the Supplementary Table 1.

The radiocarbon date information is in multiple places: 14C dates are in Supplementary Table S1, 14C date accessions are in the methods (lines 342-345), and calibrated dates are

in the methods and Table 1. Please locate these in one place, preferably Table 1. How were these dates calibrated?

We now added the calibration information in the Materials and Methods section “AMS radiocarbon dating” at lines 422-427. Briefly, the IntCal13 calibration curve in OxCal v4.2 was used to calibrate radiocarbon ages estimated by TÜBİTAK-MAM and Beta Analytic Inc. Following the Reviewer’s suggestion, we joined all C14 information (calibrated dates, direct dates in BP, and lab codes) in Supplementary Table 1.

To enable replication, the authors should list their 20,000 SNPs (line 162) in a supplementary file, including genomic coordinates and whether they are transversions, mitochondrial, or domestication-related.

We thank the Reviewer for this suggestion. We added this list as Supplementary File 1, which is now mentioned in the Materials and Methods section “Hybridization capture” at line 467.

Figure 3 and supplementary figure 5a: please re-plot these on just the first two eigenvectors, as it is impossible to visualize a 3D plot in a 2D image. In supplementary figure 5a, consider making the legend larger for readability.

We thank the Reviewer for these suggestions. Accordingly, we changed Figure 3 (now Figure 2) to include PC1 and PC2 (see **Figure R4** below), and we provide a plot of PC1 vs PC3 as Supplementary Figure 9. We likewise changed Supplementary Figures 5 and 15. In addition, we increased the font size in Supplementary Figure 5.

Figure R4. Genomic variation of modern breeds, Anatolian Neolithic sheep and Obishir V¹⁷. The first and second components of the PCA were calculated using genotypes of 18 modern sheep breeds. The four Anatolian Neolithic sheep individuals' genotypes (black triangles) and three Obishir V sheep individuals' genotypes (grey circles) were projected on these 2 components. Parentheses indicate the proportion of variance explained.

Figure 4 is missing the data from the ancient individuals (ULU, TEP)... I assume these are similar to NSP and FIN?

This plot (now Figure 3) was produced to represent the genetic similarity (measured by the outgroup f_3 -statistic) between modern-day breeds, and ANS as a *population* (i.e. pooled allele frequencies of the four individuals). We now also added the same plot produced for each ANS *individual*, as shown below **Figure R5** (Supplementary Figure 13).

We explain that the ANS data was calculated as a population, and refer to Supplementary Figure 13, in the legend of the current Figure 3.

Figure R5. Outgroup f_3 -statistics between each ANS individual and modern-day breeds. Outgroup f_3 -statistics were calculated as $f_3(\text{Argali}; \text{ANS}, \text{Modern})$ (Supplementary File 3 Sheet B). The breed name abbreviations are listed in Supplementary Table 3. Error bars indicate ± 1 standard error from the mean.

Methods: there is a lot of overlap between the methods in the main text and in the supplement (e.g. lines 382-386 and 396-402 in the main text are highly similar to the supplementary text). This makes the text confusing, as some details are in either one set of text or the other. To aid the reader, I therefore recommend that the authors combine the two methods sections into one in the main text.

We thank the Reviewer for this suggestion. We now compiled all the Materials and Methods sections in the main text, which we hope is easier to follow now.

36 libraries were constructed (line 368) from 36 samples (SI line 401). However, only 29 are listed in supplementary table S8. Why the discrepancy?

We apologize for this mistake and thank the Reviewer for pointing this out. It is not 36 but 29 libraries. These are listed in Supplementary Table 8 (now Supplementary Table 2).

State what read lengths and whether paired-end chemistry was used for both the HiSeq-4000 (SI line 402) and HiSeq-X (SI line 410) sequencing.

We thank the Reviewer for this suggestion. We added these details to the Materials and Methods at lines 460 and 476. Briefly, we sequenced paired-end with a 2X150 setup using HiSeq 4000 and also HiSeq X SBS chemistry.

Check the Supplementary Figure and Supplementary Table calls - many are out of order and/or mis-assigned (e.g. lines 121, 255, SI lines 94, 463)

We apologize for this confusion. We now reordered the figures and tables throughout the manuscript.

Minor:

line 26: define 'BCE'.

Added now at line 33.

line 29: specify that these are nuclear SNP data.

Added at now at line 23.

line 38: state when the Neolithic transition took place.

Added now at line 33.

line 75: are these 'local stocks' referring to within Europe or across the whole distribution of domesticated sheep?

We clarified this by adding “possibly across the globe” at line 74.

line 75-76: is this tMRCA based on mitochondrial DNA?

We clarified this by adding “using haplotype sharing information” at line 74

line 76: approximately how far back in time are 800 sheep generations?

We clarified this by adding “about 3200 years” at line 76 based on Ezard et al.¹⁸.

line 85: earlier in the intro, give the age range of the Neolithic.

Added now at line 33.

lines 88-89: give exact number of bones analyzed and the age ranges of these within the 'early Holocene'.

Added now at line 89-90.

lines 89-90: give the ages of the Epipaleolithic and Chalcolithic in Anatolia. It may be helpful to give these in the Figure 2 headers, and then refer to this figure.

Added at lines 91 and 92 and now provided in Figure 4 (past Figure 2).

line 95 and throughout: check order of supplementary tables.

We now correctly ordered the supplementary tables - we apologize for the confusion.

line 126: state that these values are Shannon diversity indices.

We removed these analyses and results from the manuscript.

line 127: change 'haplotypes' to 'sequences'.

This sentence is completely removed.

line 143-144: what evidence is there to suggest that lineages were introduced from the region east of the Fertile Crescent?

We now removed this speculation about possible admixture in the section presenting mtDNA results. In the Discussion, we now write “Hence, we hypothesize that central and west Anatolian Neolithic sheep were likely directly related to Europe's first domestic breeds, but the continent’s domestic sheep gene pool was strongly remolded by subsequent gene flow events of Asian and/or African origin” in the main text at lines 402-405. This is based on our nuclear DNA analysis results: a) higher affinity of Anatolian Neolithic sheep to modern European breeds, b) higher affinity of Kyrgyz Neolithic sheep to modern Asian breeds, c) lack of affinity of any modern European breed to any Neolithic sheep group over other modern breeds. Because Europe is believed to have lacked a wild sheep population by the early Holocene, post-Neolithic changes involving non-European populations appears plausible. In addition, the basis of this speculation we also articulate in the Supplementary Note1 at lines 268-292.

line 156: correct '%2'.

Corrected at line 107.

line 166-167: state here that 10 bp were trimmed from either end of the reads.

Added at line 116-117.

lines 217-218, 227-228: repetitive definitions - only need to define once.

We removed repetitive definitions.

line 225: change 'demographic histories' to 'admixture histories', as other aspects of demography (e.g. population size) were not investigated.

We removed this sentence.

line 247: change 'strains' to 'individuals'.

Changed now at line 228.

line 257: rephrase 'chose' to 'grouped with' or similar.

Changed now at line 259.

line 272: change 'we are witnessing' to the past tense.

We removed this part in the current version, following recommendation by one of the Reviewers to dampen our discussion related to domestication.

line 358: specify which sample.

In the first submission, we had wrongly assumed that haplogroup B could be misassigned A due to recurrent PMD at diagnostic positions; we had thus removed these four haplogroup A cases from the sample, as we had mentioned in the Methods. We now realise, however, that the possibility of two or more PMD-induced transitions at the same position in two or three fragments is extremely low. Hence, in the current version we did not remove any samples.

Figure 1 and supplementary figure 1a: enlarge the text on the figure, as it is currently difficult to read. If this makes the figure too busy, consider numbering the sites in red on the figure and defining these in the figure legend.

We thank the Reviewer for this suggestion. We redraw Figure 1 (see below **Figure R6**) and Supplementary Figure 1 (below **Figure R7**).

Figure R6. Map of archaeological sites. Geographic origins of samples from six Anatolian archaeological sites studied in the present study (see Supplementary Note 2 for details) and one Kyrgyzstan archaeological site from Taylor et al.¹⁷ are shown in black. Four additional Neolithic Anatolian sites relevant for sheep domestication discussed in the text are shown in red. Putative domestication center is depicted by the shaded area (adopted from Zeder¹⁹).

Figure R7. Geographic map of modern and ancient sheep samples in this study. a) Geographic locations of modern breeds (in black), ancient Anatolian sheep individuals (red) and ancient Kyrgyzstan, Obishir sheep individuals (red) used in mitochondrial DNA analyses. b) Geographic locations of modern breeds (black) and ancient individuals (red) used in genomic analyses. For population abbreviations see Supplementary Table 3.

SI line 144: specify that these are 'captured libraries'.

They include merged libraries. This is now mentioned in the legend of Supplementary Figure 7.

SI lines 183, 184, 253, 279: change 'kms' to 'km'

Corrected now throughout the text.

SI line 193-194: use 'BCE' to be consistent with the main text.

Corrected now throughout the text.

SI line 205: define 'NISP'.

Defined now at line 378.

SI lines 226, 229, 236, 241, 260, 288, 313, 444: check for typos.

Thank you. All typos corrected.

SI line 227: which sample?

Information added to the line 410.

SI lines 298-299: change to decimal point.

Changed now at line 500.

SI lines 366: change 'sandblasting' to 'abrasion'.

Changed now at main text line 447.

SI lines 378: what brand and company of Taq Polymerase?

Information added to the main text lines 589- 590.

SI lines 427-428: how were these reads removed?

Corrected now at main text line 501.

SI line 489: change 'dataset were merged with' to 'datasets were merged'.

Changed now at main text line 558.

Supplementary table 1: what is 'depo' in the lab ID names?

'Depo' means repository in Turkish. It refers to the samples obtained from the excavation's repository. This information is also added to the legend of Supplementary Table 1 legend.

Supplementary table 8: There are five libraries highlighted with an asterisk, but the legend states 'n=4'.

This typo in the legend of Supplementary Table 8 (now Supplementary Table 2) is corrected.

State whether the 'number of reads mapping' is after duplicate removal and/or filtering for mismatches.

The statistics reported were calculated using reads after duplicate removal and quality filtering, except for the endogenous proportion values (reported in Supplementary Table 2). Which read set was used is now indicated explicitly in the legends of Table 1, Supplementary Table 2, Supplementary Figure 8 (sex identification).

Supplementary figure 3: these lines do not refer to T, A, G, and C. Blue is C>T, and red is G>A.

Thank you, the legend is now corrected.

Supplementary figure 9: are these outgroup- or admixture-f3 statistics?

They are outgroup f3 statistics. It is now mentioned in the title of the figure (now Supplementary Figure 14).

Supplementary figure 10: on the x-axis, suggest annotating whether populations are European, Asian, or African.

Origins of the modern breeds are now added to the legend of Supplementary Figure 10.

We very much thank the Reviewer for their meticulous revision and contribution to our manuscript.

Reviewer #3 (Remarks to the Author):

This paper reports a substantial number of mtDNA control region sequences from ancient sheep across Central and Western Anatolia, spanning from pre-Neolithic to Chalcolithic time periods. This data suggests i) a reduction in haplotype diversity within haplogroup B following c. 7,500 BCE, and (ii) an increase in haplogroup diversity from 7,000 BCE to 5,500 BCE, a trend which apparently continues to modern day breeds in the region. The paper also presents shotgun and capture-enrichment data for four sheep from the approx. 7,000-6,000 BCE era, which they use in concert with modern sheep breed SNP data to explore their relationship with later domestic populations. The key arguments made from this data are (i) Anatolian Neolithic Sheep (ANS) show more shared drift with European breeds than non-European, (ii) in particular North/Central European breeds, and that (iii) there is a degree of discontinuity between ANS and all modern sheep breeds (with technical caveats); these three points are linked to later a spread of Asian breeds prior to modern breed differentiation.

This paper represents a substantial step forward in our understanding of the genetics of ancient sheep populations in late Epipaleolithic-Neolithic-early Chalcolithic Anatolia. To my knowledge this is the first ancient mtDNA data from Anatolian sheep during the crucial periods encompassing the onset and intensification of management. Given the importance of this region for the development and spread of sheep herding, the potential importance of this paper and resource is considerable.

The authors of this paper should be commended on the clarity of writing that characterized the majority of the text and supplementary material; it was well structured and flowed well, and would help its appeal to a broad audience as well as a more specialized one. In addition there has clearly been a substantial amount of work performed to process the number of archaeological material of this data set, and represents an important genetic resource for those studying the history of animal domestication i.e. zoo-archaeologists and archaeo-genetics. Finally I was pleased with the extent of work apparent in the supplementary texts, from methods to the zoo/archaeological context of the sites the samples are drawn from.

We thank the Reviewer for their kind appreciation.

However the study has limitations which were not addressed satisfactorily in the text as it stands. The use of a novel targeted enrichment probe set to generate the bulk of ancient nuclear data, the reliance of a modern SNPchip reference set for many tests, and finally the inclusion of transitions in their working SNP set, are all of concern. Combined with the low (relative to other ancient studies) number of called SNPs in their four enriched samples, data set quality stands out as a major technical limitation of the study.

We thank the Reviewer for sharing these concerns. We address the points about data quality below:

- We agree that relying on SNP capture data and using a modern SNP dataset together may lead to ascertainment bias and limit our analyses, e.g., prohibiting the use of measures such as theta or F_{ST} . Meanwhile, the analyses we rely heavily upon f_3 and D -statistics, which are largely robust to ascertainment bias²⁰. We now note this point in the main text on lines 124-125.
- We had adopted the strategy of trimming read ends to avoid the influence of postmortem damage on the data and thus to include transitions in order to increase sample size with respect to SNP numbers. That said, we agree that controlling for possible confounding of postmortem damage would be desirable. Thus, in this version of the study we generated a second dataset including transversion SNPs only and repeated the main analyses with this set (**Figure R8 below**; Supplementary Figure 15). The results were fully consistent with those obtained with the original dataset.
- We realized an error in our earlier SNP calling pipeline that was caused by genome version differences. Correcting this doubled our SNP list size (Table 1).

Figure R8. Principal components analysis and outgroup f_3 plots of Anatolian Neolithic sheep and modern sheep breeds calculated using transversion SNPs only. The panels a) and b) were produced in the same way as Figures 3 and 4, respectively. Note that transversion type SNPs are not expected to be confounded by postmortem damage.

- To fully address the Reviewer's criticism, we also considered another potential source of bias: reference genome bias²¹. This could be relevant as our data is low coverage, and the reference genome oviAri3 is based on a European-related breed, which could lead our ancient genomes appear artificially similar to the European breeds. To address this we generated a third dataset version using an unpublished approach, where we masked the reference genome at polymorphic sites, remapped the data on the masked genomes to generate bias-free BAM files, and called genotypes using this data. The analyses we conducted using this data again recapitulated our original results (**Figure R9 and Figure R10** below). We did not include these results in the study because the approach we used is not published yet.

Figure R9. PCA calculated using data generated with the masked reference genome.

Figure R10. Outgroup f_3 -statistics calculated using data generated with the masked reference genome.

In summary, despite limitations of our dataset, the observation that Neolithic Anatolian sheep are genetically closer to modern-day European breeds than Asian breeds, appears robust to ascertainment bias, SNP choice, or possible reference bias effects.

Further, in the current version of the manuscript we added recently published OBI¹⁷ (Neolithic/Bronze Age Kyrgyz) sheep data. These show a different behaviour than ANS in their relationship with present-day breeds, such that they are genetically closer to present-day Asian breeds. This observation again suggests that ascertainment or reference biases may be of limited concern overall.

Additionally, as presented the text and figures are not clear on details and decisions which are key to assessing the strength of the results, and which underlying the conclusions.

We revised the manuscript thoroughly to increase the clarity, and also dampened the discussion on results which are weakly supported, as suggested by the reviewers.

Finally, several of the conclusions (eg line 329 “probably locally domesticated”, line 32 “Asian contribution to south European breeds”) are over-reaching given the quality of the data.

Although this paper does represent an important step in expanding our understanding of the sheep domestication history, the technical issues must be addressed before publication, and the strength of the conclusions amended to be more conservative. Specifically, a local domestication in Central Anatolia being captured from a small number of mitochondria sequences from a limited number of sites, and the European-Asian Bronze Age admixture conclusion, are at best weakly supported from the data here, and lean substantially on the weight of previously published studies.

We agree with the Reviewer, and we now removed our conclusions about local domestication in central Anatolia based on genetic data.

In the main text, we limited the presentation of mtDNA analysis results to those supporting our results based on nuclear data, i.e. the high affinity ANS shows to present-day European breeds, and the differentiation we observe between ANS and Asian breeds, both ancient and present-day.

We discuss the possibility of local domestication based on archaeological evidence in the Supplementary Note 2 at lines 386-417, where we also refer to some of the mtDNA results, but with a strong emphasis on our limited sample size and lack of power, paying utmost care to avoid overinterpretation.

Please note that this review was performed without access to the outgroup f3 statistics quoted in Lines 227-229, which were not present in the Supplementary File 1 provided.

We sincerely apologize for this omission. We have now submitted the Supplementary Files along with the revised version of the manuscript.

Major Comments

Figure 2 of the paper presents issues with interpreting the underlying mtDNA data. The decision to not differentiate between sites within the two major regions (Central and Western Anatolia) is problematic, as it does not clearly indicate the uneven distribution of sequences from the various sites and regions. This is crucial given that the paper is arguing that the data captures a transition from wild to managed sheep - if this signal was being driven by just one site, the figure as stands would not reflect this. I note that the figure text highlights this - “Note that different periods and haplogroups are not homogenously...” - but this is not sufficient for a main figure.

We thank the Reviewer for this point. We now reorganized the figure as suggested (**Figure R1** above, now Figure 4 in main text). Additionally, we decided to remove haplotype-related analysis from the manuscript, and accordingly to simplify the figure and present only haplogroup information from each site. The main message is the prevalence of haplogroup B in ANS and increase in haplogroup A in present-day central Anatolia.

We note that the lack of data from west Anatolia before 7000 BCE owes to a lack of Neolithic settlements in the region prior to that time.

Additionally, this text contains a major issue: “while 7500-7000 BCE by Canhasan III and Tepechik-Çiftlik”. In Figure 2 itself, this period is represented by three HPG B sequences of the same haplotype. When Supplementary Table 2 is examined, the three B sequences are all in fact from Canhasan III (CH1, CH2, CH3).

We thank the Reviewer for pointing out this mistake. We hope that the current version of the figure is clear (Figure 4, also **Figure R1** above).

Although some sequences from Tepechik-Çiftlik do have (estimated and C14) ages which partially overlap the 7,000 BCE boundary, they appear to be grouped in the 7000-6000 group in Figure 2, and the associated text is somewhat misleading. The Figure should be restructured in order to express the archaeological origin of the ancient samples here.

The figure is now corrected and reformatted as suggested.

The text should specify how samples are placed in the different periods when their ages straddle the transition.

Thank you. An explanation is added to the legend of the figure (current Figure 4, and **Figure R1** above): ‘For the samples with ages straddling two periods, we took the average of their age range to place it within an archaeological period.’

Furthermore, Figure 2 does not include mtDNA haplogroups identified by NGS, not PCR - specifically the HPG sequence from Tepechik-Çiftlik (TEP83; Table 1). All other samples from Table 1 were included in the PCR-based sequencing and are presented in Figure 2. Although the NGS data may be insufficient for accurate haplotyping, it appears to be sufficient for haplogroup assignment (see comment below on NGS mtDNA methodology). As TEP83 represents the only HPG A sequence from their data, it is of interest to many readers and bolsters the paper’s argument for an increase in haplogroup

diversity post 7000 BCE. The sample - at least the presence of HPG A in the PrePottery Neolithic - should be indicated in Figure 2.

We thank the Reviewer for this valuable suggestion. We added NGS-based haplogroups (TEP03, TEP62, TEP83, ULU26, ULU31) to the figure (now Figure 4) and to Supplementary Table 1. The presence of haplogroup A within the period of 7500-7000 BCE was emphasized in the main text at lines 323-326.

The modern sequences in Figure 2 reflect the samples chosen for haplogroup comparison, rather than an accurate representation of haplogroup frequencies in modern Turkey (Supplementary Table 2). For example the central Turkish breed Akkaraman has 52% HPG B in Sup Table 2, but according to Figure 2 sheep from the region have $4/21 = \sim 19\%$ freq of HPG B. This is quite misleading, over-estimating the apparent change in haplogroup frequencies, and the apparent difference in changes in mtDNA diversity between Western and Central Anatolia. This subsequently affects Haplogroup diversity estimates (Sup Table 5). The authors should have downsampled the Central Anatolian 144bp sequences to reflect their actual present day frequencies and repeat the haplogroup diversity analysis.

We thank the Reviewer for this point and apologize for the confusion.

We now show the full present-day Anatolian breed mtDNA haplogroup frequency data in the figure (Figure 4 in the current version, and **Figure R1** above), consistent with that presented in Supplementary Table 6 of the current version. (The reason for the discrepancy in the earlier version was that we had presented the data from which we had haplotype information, which as a biased subset of the full data. Now we have removed all haplotype-based analyses from the study.)

We also removed haplogroup diversity comparisons as we decided the sample size was too small to reach clear conclusions. We only compare haplogroup frequencies using the Fisher's exact test and reporting effect size by using Cohen's w (Supplementary Table 10).

High data quantity and quality is a luxury rare in ancient DNA, and inferences are often limited by this. This paper does present a substantive jump forward in the genomic data from ancient sheep populations, but SNP coverage is still low (3,294-10,484). Given this, it is essential that uncertainty from standard errors and number of snps in each test be presented for all tests, as they are not in the present text or supplementary materials/supplementary file 2.

Thank you. We now added the standard errors and SNP numbers in all tables in Supplementary File 2 and Supplementary File 3.

Additionally, the D statistic tests quoted lines 216-225 should be included in the supplementary files, particularly with the number of ABBA and BABA sites for each, as the lack of statistical significance could be a consequence of low SNP number.

We thank the Reviewer for this suggestion. We now provided the ABBA and BABA numbers, along with the SNP numbers in Supplementary Files 2 and 4. In the test involving the smallest sample sizes (SNP numbers), we still have >3000 SNPs and >200 ABBA and BABA counts. It therefore is likely that any genetic structure in Anatolian Neolithic sheep was weak and the lack of differentiation was not simply a power issue. We now note this at lines 205-206 in the main text.

Meanwhile, in the newly added *D*-tests involving both OBI (ancient Kyrgyz sheep) and ANS, we have SNPs between 1553-2253 and ABBA and BABA sites between 107-221. Here we do have a power issue, which we acknowledge at lines 237-239 in the main text.

Regarding the modern breed data, the text is often confusing about whether it is referring to geographic regional groupings or groupings from the original source (Kijas, 2012). For example, does “Eastern Mediterranean strains” refer to the Leccese breed? Is Sakiz an “Eastern Mediterranean“ breed, while being grouped in “SW Asia” in Sup Table 7? In lines 248-253, are the “central”, “northern”, and “south” European breeds referring to the groupings in Sup Table 7? Similarly, lines 311 mention “Middle Eastern”, “south” and “central/north European” sheep.” The authors should be clear in the text and Sup Table about which grouping they are using, ideally delineating their terminology in Sup Table 7.

We thank the Reviewer for this helpful suggestion. We now clarified geographical classifications of the breeds and present them in Supplementary Table 3.

Following both comments 4 and 6, key statistical tests are lacking from Supplementary File 1. The authors state (lines 227-229) that outgroup f3 statistics using Argali sheep as the outgroup were presented in this file, but these tests were not in the file provided by the authors. Given that these statistics underlie a central result and figure (Figure 4), these tests must be provided to readers. Additionally, being able to directly assess the standard errors associated with each test would be very useful for readers, given how similar the median values are for the European vs non European shared drift (albeit statistically distinct based on the Mann Whitney U-test, lines 241-243).

We are sorry for the omission, and thank the Reviewer for the suggestion. In the current version, Supplemental Files 3 and 5, which present f3-statistics results also include standard errors.

We also present now Supplementary Figure 13 (**Figure R5** above), which shows $f_3(\text{Argali}; \text{ANS}, \text{Modern})$ values with their standard errors.

Finally, why did the authors not employ the same statistical test for determining differences in the shared drift with North/Central European breeds vs South European?

We now removed the analyses regarding differences between North/Central and South European breeds because we decided the signal was too weak.

The major comments above (and some minor below) lead me to conclude that the present study overstates their data's power to support one of the key conclusions, line 31-33: "Our results indicate that Asian contribution to south European breeds in the post-Neolithic era, possibly during the Bronze Age." The key three pieces of evidence supporting this have issues. As discussed, the D statistic bias towards allele sharing between ANS and North/Central European breeds may be the product of reference bias. ADMIXTURE results (see below) do not confidently establish a difference in ancestry similarity between North, Central, and Southern European breeds, so it is selective to state that ANS show an ancestry profile most similar to N/C European breeds (lines 253-254).

We agree with the Reviewer. We now removed our previous conclusions regarding 'Asian contribution to south European breeds in the post-Neolithic era, possibly during the Bronze Age.' Overall, we do not attempt to distinguish between N and S European breeds in the current version of the study.

And finally, as the authors acknowledge (lines 258-259), the D test indicating greater allele sharing among modern breeds than with any modern breed and ANS is possibly a technical issue, which could be the persistence of deamination error in the data (see below).

This is a valid point, but we believe the impact of postmortem damage-induced deamination is limited, (1) having repeated the D -statistics with the transversion only dataset and having obtained similar results (Supplementary File 4), (2) observing a similar pattern of modern-modern affinity relative to ancient when we tested $D(\text{Argali}, \text{Modern}_1; \text{Modern}_2, \text{OBI})$, given that the OBI data was produced using a UDG-based protocol and without SNP capture. Thus, while remaining cautious, we are inclined to believe that there could be a biological cause for these D -statistics results. This we now explain in the main text at line 386-405.

The D statistic tests using just modern breeds are not sufficient (lines 308-313), as we are not presented the number of tests which do not support the affinity between Middle Eastern and South European breeds, and authors do not raise a very plausible alternative hypothesis - that the two regions, being geographically close, experienced greater trade

and gene flow. As such, unless the authors present transversion-only results which also support this conclusion, it is difficult to conclude that the data supports this major conclusion.

We now completely removed the analyses regarding differences between North/Central and South European breeds.

Following from the previous comment, the authors introduce some hypotheses that their data does support, even without the issues discussed. Specifically, the Bronze Age is not discussed in any detail outside of the introduction of wooly breeds to Europe in the Introduction (31-33). If the genomic data was from Europe and/or Bronze Age contexts, such inferences would be fair, but currently the data from Neolithic Anatolia is too weak to support this kind of speculation in the Abstract.

We agree with the Reviewer with her/his criticism and now corrected this claim as post-Neolithic contribution instead of Bronze Age.

3D PCAs (Figure 3) are very difficult to interpret on a 2D plane. At the very least they should consider pairwise plots of PCs presented in the SI, if not replacing Figure 3 with a 2D equivalent. In addition, transition error could be driving the “relative central location” (line 214) of the ANS samples in the PCA. The PCA should be repeated using transversions-only and presented in the SI. Note that for this and the previous comment, transversions-only may be too few sites for robust population analysis.

We thank the Reviewer for this suggestion. We now replaced the 3D PCA in Figure 3 (now Figure 2, and **Figure R4** above) with a 2D PCA. We also now present a transversions-only PCA (Supplementary Figure 15a, **Figure R8a** above).

The authors should consider determining the frequency at which ANS samples show the damage allele at transition sites in their data, compared to individual modern breeds or all modern breeds.

We thank the Reviewer for this valuable suggestion. We now present this result as Supplementary Figure 3 (and **Figure R12** below). This shows the observed frequencies of possible postmortem damage-resulting alleles (T and A) at transition sites in the modern breed dataset and in ANS individuals' data. This analysis revealed an excess of T alleles in 3/4 ANS individuals, but no consistent excess of A alleles at A/G sites (note that the higher noise in A or T frequency in ancient samples is arises due to our lower sample sizes). Thus, we cannot fully rule out a low degree of residual postmortem damage, for which reason we repeated the population genomic analyses also using transversion SNPs (**Figure R8** above; Supplementary Figure 15).

Figure R12. Frequency of damaged alleles at transition sites in Anatolian Neolithic sheep vs. modern-day breed genotypes. For each ancient and modern individual, proportion of A at A/G polymorphisms, and proportion of T at C/T polymorphisms are shown. Blue points show modern-day breeds (n=NNN), while ANS are shown in red (see legend).

The authors report unusually long fragments for their ancient data. Although the issues raised above may confound the results, a few additional steps would aid in firmly establishing their authenticity. mtDNA haplotypes as determined from long vs short reads should be identical, once missingness and damage is accounted for; the authors check these against each other. Additionally, the size distribution of aligned fragments should be presented for each sample; unimodality would support an ancient origin of the long reads.

We thank the Reviewer for this helpful suggestion. We determined mtDNA haplotypes with reads shorter than 70bp and with reads longer than 100 bp, and identified the same haplogroups with both data (**Figure R13** below). We did not include these results in the manuscript as we decided to overall dampen our results regarding mtDNA haplogroups, as more than one of our Reviewers suggested. Instead, we performed an analogous analysis using outgroup f_3 results, where we calculated $f_3(Outgroup; ANS_i, Modern_breed_j)$ values using reads <70bp and reads >100 bp per ANS individual, and calculated the correlation across these values (Supplementary Figure 7, **Figure R14** below).

Figure R13: NJ trees of eight ancient mitogenomes, 10 modern genomes representing each haplogroup by 2 samples¹ and *Ovis vignei* as outgroup¹. a) NJ tree with reads shorter than 70 bp b) NJ tree with reads longer than 100 bp. Mitogenomes shown by straight arrows belong to Anatolian Neolithic Sheep samples (n=5) presented in this study and mitogenomes shown by dashed arrows belong to Obishir V¹⁷ (n=3).

Figure R14. Comparison of outgroup f_3 -statistics calculated for all (unfiltered), short (< 70 bp) and long (> 100 bp) molecules of each ANS individuals. We calculated $f_3(\text{Argali}; \text{ANS}_i; \text{Modern})$ using different molecule sets in the merged (shotgun and capture joined) libraries of each ANS individual (TEP03, TEP62, TEP83 and ULU31). In the scatter plots presented in the lower triangle of each panel, each point represents an f_3 value calculated as $f_3(\text{Argali}; \text{ANS}_i; \text{Modern})$ where *Modern* is a modern breed, and the two axes represent f_3 values calculated from different molecule sets (all, short, or long). The red line represents linear regression. Panels in the upper triangle of each panel show the Spearman correlation coefficients for the corresponding scatter plots; asterisks indicate significance of the Spearman test. The diagonal in each set represents the distribution of f_3 values calculated for all reads (unfiltered), short reads <70 bp and long reads >100 bp. The figure was generated by R (v.3.5)¹⁵ library ggplot2²² in RStudio (v.1.3)²³.

The supplementary methods do not mention any control experiments e.g. for the DNA extraction or PCR amplification. Given that 60/74 of PCR mtDNA sequences from Figure 2 have the same haplotype, the same haplotype “most widespread today” (line 137), negative controls would be vital in ruling out modern contamination. Could the authors clarify if these were performed. Similarly for Supplementary Table 8, there is no mention of negative control libraries included in the NGS and enrichment. Could the authors clarify if these were performed, preferably including the sequencing results of the control libraries e.g. proportion of reads aligning to sheep.

We thank the Reviewer for pointing out this omission. We included negative controls in every step of the whole genome library preparations and also in mtDNA amplifications. We now added a new section named ‘Procedures for contamination prevention and detection’ to Material and Methods, lines 428-445. Briefly, in NGS experiments, we quantified the negative controls included both during DNA extraction and Illumina library preparation using Agilent2100 Bioanalyzer; we found no trace of contamination in any of the experiments. In mtDNA experiments, negative controls were checked on an agarose gel along with the amplification reactions for ancient samples. Our agarose gel runs likewise did not provide any evidence for modern DNA contamination.

Line 101, “each likely from distinct individuals according to context (Supplementary Material and Methods)” - from the supplementary text, I could not see any indication how samples were determined to be “likely” distinct - could the others please clarify. Inclusion of material type (talus, molar, left petrosal etc) in Supplementary Table 1 could help this.

We apologize for the missing information. We now added detailed information based on the archaeological context of the samples under each excavation’s section in Supplementary Note 2, at lines 1133-1148 (Pınarbaşı), lines 1205-1207 (Can Hasan III), lines 1246-1257 (Tepecik-Çiftlik), lines 1290-1303 (Ulucak Höyük), lines 1329-1335(Barcın Höyük).

Minor Comments

The authors should touch on the identity of the wild progenitor of domestic sheep, and other relevant Ovis species, in the introduction, as it is relevant to later statistical tests.

We now added a brief information about the wild progenitor of domestic sheep as well as its close relatives used in the later analyses, at lines 33-36 in the main text. We also added a more detailed section in Supplementary Note 1, lines 245-267.

Line 82, the authors mention “one of the possible domestication centres”. The introduction should be more specific in what these potential alternatives centres are, and perhaps

Added at lines 357-358.

Line 127 - “totally vanished” not appropriate language, change.

This expression is removed.

Line 146-147 - if the authors state that that the change in haplogroup composition “happens more subtly” in Western Anatolia, they should provide a reference to their results backing up the statement (Haplogroup diversity from Sup Table 5 etc).

We now removed this statement.

Line 148-149, “analyses of maternal lineages lend support to a domestication event in central Anatolia” - the data is also consistent with domestic sheep being introduced to the region; the presence of the later-common HPG B type in pre 7,500 BCE Pınarbaşı is not sufficient to conclude that domestication occurred in Central Anatolia.

We agree with the Reviewer, and we now removed this statement about local domestication in central Anatolia from the main text. Supplementary Note 2 elaborates on the possibility of sheep domestication in central Anatolia based on archaeological data, while also briefly referring to the mtDNA-based observations here, and paying utmost care to avoid overinterpretation.

Line 141, “our sample size is too small to exclude the presence of non-B haplogroups” - four samples is certainly too few to draw this conclusion. The authors could easily explore this e.g. if the frequencies in Central Anatolia between 7500-7000 and 7000-6000 are same, the chances of sampling a non HPG B from three sequences, or the maximum frequency a haplogroup would have to be to sample at least one in three samples at say 0.95 probability. This would contextualize the speculation that follows.

We thank the Reviewer for this helpful suggestion. The sample size does not provide power to reject the null hypothesis of no difference. We thus changed the text as follows: “[...]we observe non-B haplogroups, including haplogroup A, in our ancient sample only post-7500 BCE, during the Ceramic period (Figure 4). However, our sample size is yet too small to exclude the presence of non-B haplogroups in central Anatolia pre-7500 BCE. Whether there occurred an increase in haplogroup diversity in central and west Anatolia post-7500 BCE yet remains unclear (Figure 4; Supplementary Note 1).”

Line 159 - according to Supplementary Table 2, the calibrated age of TEP03 also extends into the 8th millennium, not just TEP62. Please correct.

We corrected this statement at lines 109-110.

Line 190-191 - four samples is far too few to make any statement on sex bias. Three out of four sheep being female would be consistent with a bias towards males as well, given the level of statistical power. I recommend removing reference here to sex bias.

We removed references to the sex bias.

Line 224, “from different periods” - are the authors referring to the periods as defined in text i.e. 7500-7000 BCE and 7000-6000 BCE? As the four samples all have C14 ages which predominantly lie within the 7000-6000 BCE period.

We now removed this expression.

Line 253-254 - “ADMIXTURE...indicated similarity between ancestry components in AND and north and central European breeds” - visually assessing all Ks, the ANS sheep also have similar profiles to south Europeans breeds e.g. LEC; European breeds appear to have very similar profiles. Differences present are subtle at best and the authors do not present error bars on ancestry fractions in the ANS samples to assess ADMIXTURE results (low SNP number, transitions could produce high variation between runs). From this analysis alone, it is not possible to say that they show more similar profiles to any one European regional group, but could state a greater similarity in profiles to European breeds than non-European. The authors should amend the text to reflect this, and give some indication of the uncertainty of ADMIXTURE results on their data.

We agree with the Reviewer. ANS did indeed show higher similarity to south European breeds in ADMIXTURE (as opposed to the trend observed in *D*-tests). We believe the reason was stronger drift in north European breeds like Finnsheep (ADMIXTURE components will reflect drift, but *D*-test results not). However, because the north vs. south Europe comparison results were too weak, we decided to remove them altogether.

We did follow the Reviewer’s recommendation to use ADMIXTURE results in Europe vs. Asia comparison, using a correlation-based approach. This is now presented in Supplementary Figure 11 and Supplementary Figure 12 (**Figure R15** and **Figure R16** below).

Figure R15. Correlation coefficients for 18 breeds, calculated on ADMIXTURE components. We calculated ancestral component values from ADMIXTURE analysis with $K=8$, for each of $n=18$ modern-day breeds, using the average value across each individual within the sample. We then calculated the Spearman correlation coefficient between the average ancestral component values of each modern-day breed and each ANS individual. Red and blue represent high and low correlation, respectively.

Figure R16. ANS affinity to European and non-European breeds based on ADMIXTURE ancestry components. The boxplots show the distributions of Spearman correlation coefficients calculated between the weights of ADMIXTURE ancestry components of each ANS sheep and those of European breeds (orange) (MER, BOS, CHU, COM, ERS, FIN, LEC, NSP, SAB, VBS) and non-European breeds (dark cyan) (AFS, CHA, CHI, CFT, IDC, EMZ, NDZ, SKZ), with total $n=80$ and $n=64$, respectively. The Mann-Whitney U test was performed to assess the difference between the two correlation coefficient distributions. P-values are shown in the inset.

Line 270 and also line 26 - “loss of mitochondrial haplotype diversity” - the authors should be clear that this loss is specifically HPG B haplotype diversity. This applies to the entire paper; the authors often refer to “haplotype diversity” (e.g. line 99 in SI); if the haplotype diversity estimates are based solely on HPG B sequences, the text should indicate as such and not solely in Supplementary Table 5 (it is not clear from main text or SI methods).

We now removed haplotype diversity analyses from the main text.

Line 275-276 - acknowledging that the authors are careful just raise the possibility of a local domestication in Central Anatolia based on the presence of the later-ubiquitous lineage of HPG B, this is based from a single contextually-dated sample, not C14 dated. This should be indicated in the text.

We now removed the discussion about the possibility of local domestication in central Anatolia from the main text. Supplementary Note 2 elaborates on the possibility of sheep domestication in central Anatolia based on archaeological data, while also briefly referring to the mtDNA-based observations here, and paying utmost care to avoid overinterpretation.

Lines 298-299 - the strength of these conclusions may be affected by the technical issues discussed above, and the data may not have the power to support this.

Line 328-329, “central Anatolian wild sheep were probably locally domesticated” - again this inference is drawn from a single undated haplotype from Pınarbaşı. I do not think the current genetic data is strong enough to come to this conclusion. An introduction of domestic sheep to the region is also quite possible, as a single haplotype sequence is not sufficient to discount it. The discussed lines should be toned down or alternatives emphasized more, as the data is not sufficient to directly support these conclusions.

We concur with the Reviewer, and removed our discussions about the possibility of local domestication based on haplotypes from the main text.

The authors do not specify how Illumina data was aligned to a mitochondrial reference; this should be added to the supplementary methods.

We apologize for the omission. We aligned to the *Ovis aries* (v 3.1) reference genome containing 26 autosomes, MT and chromosome X, using the BWA aln²⁴ algorithm. This is now explained in the Materials and Methods section at lines 498-499.

On line 358 the authors refer to inconsistent mtDNA genotypes for “one individual” possibly due to postmortem damage. Could the authors elaborate, and indicate which individual the inconsistent genotypes were generated from.

In the first version of the manuscript, we had removed all four haplogroup A sequences identified by Sanger sequencing. The reason was as follows: we had observed that TEP83, which has both Sanger and NGS data, was assigned haplogroup A with Sanger data, but assigned haplogroup B with NGS data. We hypothesized that recurrent postmortem damage (PMD) at diagnostic positions could have caused a B->A misassignment (as HPG B carries T where HPG A carries C). We reasoned that we could not trust other haplogroup A sequences as well (given the predominance of B in the ancient sample), so we removed all cases of HPG A, and reported this in the manuscript.

However, we later realized that the real issue was an error in our code (related to the Samtools parameters chosen when analysing the NGS data). Correcting this, the inconsistency disappeared – the haplogroup of TEP83 was identified as A also with the NGS data. We further reasoned that the possibility of PMD-induced transitions at the same position in >1 fragment

is extremely low. Accordingly, we have included all samples which produced amplification in this version.

We now added information about all 180 specimens that we studied, including those that did not amplify, as well as those with inconsistent haplogroup assignment in Supplementary Table 1. We also now present our strategy used in haplotype assignment and provide information on inconsistencies in Materials and Methods lines 598-618: (i) In Sanger sequencing, we aimed to produce at least two 144 bp sequences per sample; (ii) for each sample, we compared the sequences to determine any discrepancies; (iii) in case of discrepancy we amplified and sequenced a third round, and used the consensus genotype; (iv) in samples where sufficient mitogenome data (>1500 bp coverage) was available, we also used this data to identify haplogroups, (v) in two cases (TEP03 and ULU31) where there was discrepancy between the Sanger-based assignment and mitogenome-based assignment, which appears to arise due to convergent transitions (homoplasy) at diagnostic positions, we reported the latter result. In Supplementary Note 1 lines 305-350 we provide further detailed information about these discrepancies and explain why we believe this was a genuine result caused by homoplasy, and not an artifact.

Supplementary Table 9 - in some cases it is not clear which base was taken as the ancestral base e.g. OAR3_141586525.1. Could the table indicate explicitly which base was taken as ancestral at each SNP.

Thank you. We clarified the ancestral state of each SNP at Supplementary Table 9 (now Supplementary Table 6), where we show the inferred ancestral allele in bold.

Supplementary Figure 4 - could the authors also indicate the median read lengths, which would likely be a more informative statistic in this case.

Thank you. We added median read length to Supplementary Figure 4.

Supplementary Figures 7 and 9 - please indicate on each comparison which correlation is significant. This is not clear at all for Sig Figure 9.

Thank you. We now show statistical significance of the correlation coefficients in both figures (now Supplementary Figures 7 and 14).

Supplementary File 2, although containing the tests it should (D statistics), does not have sufficient data for assessment. The number of SNP, ABBA sites, and BABA sites for each test should all be presented, ideally with the standard error (although these can be calculated from the D and Z score, it would be clearer to present all of this information

together). Finally if the authors could clearly show which of the two sets of tests each individual test is included in (i.e. either lines 244-245 or line 249), this would add clarity for readers.

Thank you for the suggestion. We now added the number of SNP's, ABBA and BABA sites to all D-statistics results (Supplementary Files 2 and 4). We now also referenced sheet number (in Supplementary File 2-3-4-5) in the main text for each D-statistic and f3-statistic.

References

1. Meadows, J. R. S., Hiendleder, S. & Kijas, J. W. Haplogroup relationships between domestic and wild sheep resolved using a mitogenome panel. *Heredity (Edinb)*. **106**, 700–706 (2011).
2. Cai, D.-W., Han, L., Zhang, X.-L., Zhou, H. & Zhu, H. DNA analysis of archaeological sheep remains from China. *J. Archaeol. Sci.* **34**, 1347–1355 (2007).
3. Demirci, S. *et al.* Mitochondrial DNA diversity of modern, ancient and wild sheep (*Ovis gmelinii anatolica*) from Turkey: New insights on the evolutionary history of sheep. *PLoS One* **8**, e81952 (2013).
4. Keis, M. *et al.* Complete mitochondrial genomes and a novel spatial genetic method reveal cryptic phylogeographical structure and migration patterns among brown bears in north-western Eurasia. *J. Biogeogr.* **40**, 915–927 (2013).
5. Larson, G. & Burger, J. A population genetics view of animal domestication. *Trends Genet.* **29**, 197–205 (2013).
6. Meadows, J. R. S., Cemal, I., Karaca, O., Gootwine, E. & Kijas, J. W. Five ovine mitochondrial lineages identified from sheep breeds of the Near East. *Genetics* **175**, 1371–1379 (2007).
7. Lv, F.-H. *et al.* Mitogenomic meta-analysis identifies two phases of migration in the history of Eastern Eurasian sheep. *Mol. Biol. Evol.* **32**, 2515–2533 (2015).
8. Luo, Y. Z. *et al.* Origin and genetic diversity of Mongolian and Chinese sheep using mitochondrial DNA D-loop sequences. *Yi Chuan Xue Bao* **32**, 1256–1265 (2005).
9. Singh, S., Kumar Jr, S., Kolte, A. P. & Kumar, S. Extensive variation and sub-structuring in lineage a mtDNA in Indian sheep: Genetic evidence for domestication of sheep in India. *PLoS One* **8**, e77858 (2013).
10. Gorkhali, N. A., Han, J. L. & Ma, Y. H. Mitochondrial DNA variation in indigenous sheep (*Ovis aries*) breeds of Nepal. *Trop. Agric. Res.* **26**, 632–641 (2015).
11. Tapio, M. *et al.* Sheep mitochondrial DNA variation in European, Caucasian, and Central Asian areas. *Mol. Biol. Evol.* **23**, 1776–1783 (2006).
12. Ćinkulov, M. *et al.* Genetic differentiation between the Old and New types of Serbian Tsigai sheep. *Genet. Sel. Evol.* **40**, 321–331 (2008).
13. Pereira, F. *et al.* Genetic signatures of a Mediterranean influence in Iberian Peninsula sheep husbandry. *Mol. Biol. Evol.* **23**, 1420–1426 (2006).
14. Skoglund, P. *et al.* Separating endogenous ancient DNA from modern day contamination in a Siberian Neandertal. *PNAS* **111**, 2229–2234 (2014).
15. R Core Team. A language and environment for statistical computing. R Foundation for Statistical Computing, Vienna, Austria. <https://www.r-project.org/>. (2021).
16. Muigai, A. W. T. & Hanotte, O. The origin of African sheep: Archaeological and genetic perspectives. *African Archaeol. Rev.* **30**, 39–50 (2013).

17. Taylor, W. T. T. *et al.* Evidence for early dispersal of domestic sheep into Central Asia. *Nat. Hum. Behav.* (2021). doi:10.1038/s41562-021-01083-y
18. Ezard, T. H. G., Côté, S. D. & Pelletier, F. Eco-evolutionary dynamics: Disentangling phenotypic, environmental and population fluctuations. *Philos. Trans. R. Soc. B Biol. Sci.* **364**, 1491–1498 (2009).
19. Zeder, M. A. Out of the fertile crescent: The dispersal of domestic livestock through Europe and Africa. in *Human Dispersal and Species Movement: From Prehistory to the Present* (eds. Boivin, N., Crassard, R. & Petraglia, M.) 261–303 (Cambridge University Press, 2017).
20. Patterson, N., Price, A. L. & Reich, D. Population structure and eigenanalysis. *PLOS Genet.* **2**, e190 (2006).
21. Günther, T. & Nettelblad, C. The presence and impact of reference bias on population genomic studies of prehistoric human populations. *PLOS Genet.* **15**, e1008302 (2019).
22. Wickham, H. *ggplot2: Elegant graphics for data analysis.* (Springer-Verlag).
23. RStudio Team. RStudio: Integrated Development Environment for R. RStudio, PBC, Boston, MA. <http://www.rstudio.com/>. (2020).
24. Li, H. & Durbin, R. Fast and accurate short read alignment with Burrows – Wheeler transform. *Bioinformatics* **25**, 1754–1760 (2009).

REVIEWERS' COMMENTS:

Reviewer #1 (Remarks to the Author):

I applaud the authors for their thorough revision of the manuscript addressing the extensive reviewer comments. The revisions definitely improved the manuscript and I have no further comments.

Reviewer #2 (Remarks to the Author):

I thank the authors for thoroughly addressing my previous comments. The manuscript is very much improved. Before I recommend the manuscript be accepted, I would like the authors to address the following comments:

L. 428-445, 594: The authors now provide details on the negative control procedures but do not state in the manuscript whether any contamination was observed. Based on the response to Reviewer three, this seems to not be the case but it is crucial that such a statement be included in the manuscript itself.

L. 33: should this age range for the Neolithic transition not be 10,000 - 8,000 BCE? The range to 6,000 BCE contradicts the ages given in Figure 4.

SI L. 43-44, SI File 6: please use the universal newick/tree format for this file, otherwise this is inaccessible to those without MEGA-X windows software. Even with MEGA-X software, I could not open the file on a mac.

SI L. 45, SI File 7: The legend states 318 mitogenomes but there are only 315 listed in SI File 7.